# TEARS: Textual Representations for Scrutable Recommendations

## Abstract

Traditional recommender systems rely on high-dimensional (latent) embeddings for modeling user-item interactions, often resulting in opaque representations that lack interpretability. Moreover, these systems offer limited control to users over their recommendations. Inspired by recent work, we introduce *TExtuAl Representations for Scrutable recommendations* (TEARS) to address these challenges. Instead of representing a user's interests through latent embeddings, TEARS encodes them in natural text, providing transparency and allowing users to edit them. To encode such preferences, we use modern LLMs to generate high-quality user summaries which we find uniquely capture user preferences. Using these summaries we take a hybrid approach where we use an optimal transport procedure to align the summaries' representations with the representation of a standard VAE for collaborative filtering. We find this approach can surpass the performance of the three popular VAE models while providing user-controllable recommendations. We further analyze the controllability of TEARS through three simulated user tasks to evaluate the effectiveness of user edits on their summaries. Our code and all user-summaries can be seen in an anonymized repository[1].

## CCS Concepts

• **Information systems** → *Retrieval models and ranking*.

## Keywords

Recommender Systems, Large Language Models

**ACM Reference Format:**
Anonymous Author(s). 2024. TEARS: Textual Representations for Scrutable Recommendations . In . ACM, New York, NY, USA, 34 pages. https://doi.org/10.1145/nnnnnnn.nnnnnnn

## 1 Introduction

Recommender systems are a crucial component of the online ecosystem, providing personalized content by modeling user preferences. Users daily rely on recommender systems that infer their preferences and surface relevant items rather than parsing through a large collection of items.

Recommender systems often employ collaborative filtering (CF) models, such as those discussed in [17, 54, 59], which are particularly effective for users with extensive interaction histories (e.g., clicks or ratings). These models derive latent *user representations*

[1]https://anonymous.4open.science/r/TEARS-176D/

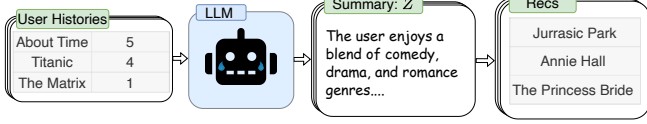

**Figure 1: General scrutable recommendations framework proposed by Radlinski et al. [40]. Our work implements this framework while also addressing its limitations.**

from observed preferences to generate recommendations. However, these representations are encoded using high-dimensional numeric vectors, which lack interpretability. Further, these CF systems offer limited control to users, who can influence them only through coarse item-level interactions, such as clicks, without understanding the precise impact of such actions on (future) recommendations.

To address these limitations, we introduce a recommender system that represents users with *natural text* summaries. Such user representations are easily understandable and directly editable [40]. Previous attempts at designing controllable recommender systems have generally restricted user profiles to broad tags or rigid templates [15]. These methods provide limited customization options, as users might find the available tags too numerous and the templates overly restrictive. Instead, text-based representations provide users with a clear view into the model's interpretation of their historical behavior (preferences) and allow them to modify these interpretations, thereby directly influencing their recommendations.

Our work draws inspiration from the framework developed by Radlinski et al. [40], illustrated in Figure 1. This framework, suggests transitioning from black-box user representations to more interpretable ones using *scrutable (natural) language* as a bottleneck to the system, similar to concept bottleneck models [26, 27, 58]. Where, they define scrutable language as being both short and clear enough for someone to review and edit directly (to impact the downstream recommendations). This approach provides several key benefits. First, it enhances transparency in the recommendation process by basing recommendations on user summaries and clarifying the system's inferred preferences. Additionally, it allows users to edit their summaries, thus giving them control over their recommended content. However, this framework assumes that text summaries can encapsulate all the information typically contained in rich numerical latents, which we find is generally not the case in practice, potentially leading to a substantial drop in performance (see Table 1).

We develop TExtuAl Representations for Scrutable Recommendations (TEARS) to obtain high-quality recommendations from scrutable user representations. TEARS uses two encoders, a regular black-box model which processes historical interactions onto numerical black-box embeddings, and another that takes scrutable, user-editable text summaries and transforms them into summary-based embeddings. We then apply an optimal transport (OT) regularizer to align these two embeddings, which are then merged using a convex combination. Changing the mixing coefficient allows the system designer or its users to guide their recommendations further. For example, users can choose recommendations based entirely on

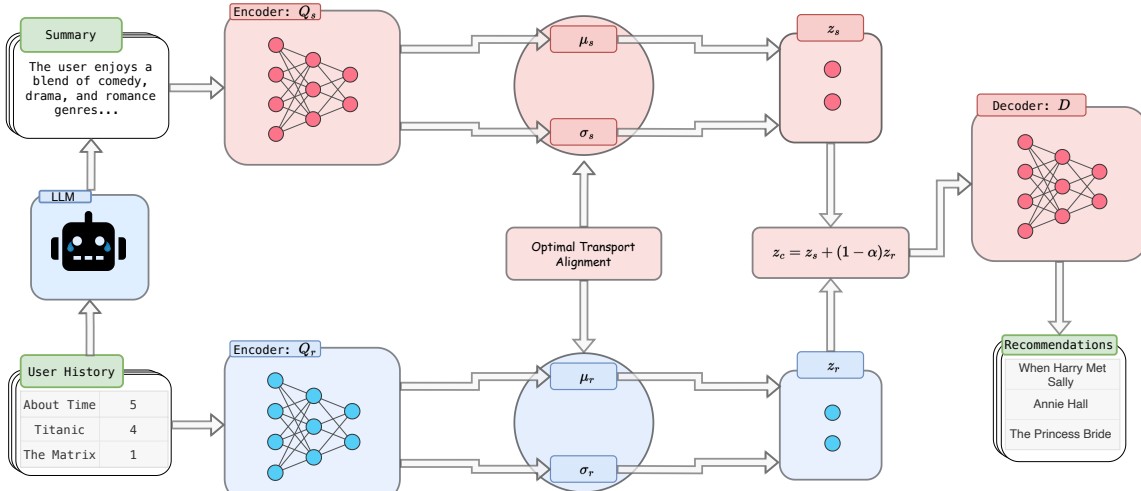

**Figure 2: We illustrate the general TEARS. TEARS produces recommendations based on a convex combination of aligned summary and black-box representations, allowing users to interpolate between transparent text-based recommendations and black-box methods. All figures in blue indicate frozen weights, while red indicates a trainable procedure.**

their user summaries, adhering to the principles of Radlinski et al. [40], opt for more black-box-based recommendations for optimal performance, or select a blend of both from text adjustments.

In our empirical evaluations, we explore three key aspects of TEARS: user summaries, recommendation performance, and controllability. We begin by testing whether modern pre-trained LLMs can generate distinctive, appropriately sized user summaries. Next, we demonstrate that aligning black-box and summary-based embeddings improves recommendation performance. Due to the lack of standard metrics for controllability, we introduce new metrics and benchmark tasks, designed to evaluate how user edits influence the system. These tasks are built around the principle that there are two primary types of user edits: large-scope and small-targeted edits, with additional changes simply being repetitions or a combination of these. For instance, we assess large changes by instructing GPT to "flip" user preferences, swapping favorite and least favorite genres, and measuring the change in the recommendations. Targeted edits are evaluated using GPT to make minor adjustments to the summary to boost the rank of a poorly ranked movie. Finally, we test an edit unique to TEARS, whereby using a mix of text and black-box embeddings, users can use short phrases to guide their recommendations. Our findings indicate that TEARS is controllable across all scenarios.

## 2 Related Work

*Scrutable Recommender Systems.* Building recommender systems using latent-variable models [48] is a common practice, but it complicates explaining the system's behavior. Although there are several works focused on explainability, most emphasize feature-based explanations [10, 37, 46, 49], or on generating *posthoc* language-based explanations for recommendations using LLMs [5, 9, 14, 28, 29]. While both approaches provide value to users and practitioners, they lack *actionability*, meaning it is not simple for users or practitioners to directly influence produced recommendations through interactions with the explanations.

Scrutable recommender systems, in contrast, have been limited in flexibility, largely relying on keyword or tag-based methods [4, 9, 10, 13, 35, 52], where users can toggle tags on or off. However, this approach can hinder controllability, as users are burdened with parsing through numerous keywords or tags, leading to significant cognitive load. Few works have explored scrutable systems through natural language user summaries. Sanner et al. [43] conducted a user study that highlighted the advantages of using user summaries to generate zero or few-shot recommendations with a frozen LLM. The study found that under specific conditions, LLMs leveraging user summaries can compete with black-box models in cold-start settings. Closest to TEARS, Ramos et al. [42] developed a scrutable model using natural language profiles for next-item prediction. However, the approach relies on user reviews, which are not always available, and only approaches the performance of black-box models. In contrast, our work does not require user reviews and employs a hybrid methodology to demonstrate that TEARS can surpass black-box model performance and maintaining scrutability.

*Large Language Models for Recommendations Systems.* Enhancing recommender systems with textual attributes is widely recognized for improving both performance and robustness [2, 23, 36, 60]. This integration generally takes one of two approaches: utilizing LLMs as standalone recommender systems and enhancing existing systems with LLM-generated text.

Studies have shown that LLM-generated text can improve recommendation quality by providing item or user-specific content [1, 50, 56]. Such approaches leverage the descriptive power of LLMs to enrich the contextual understanding of user preferences and catalog items. Meanwhile, other approaches deploy LLMs directly as recommender systems. This can be done either in a zero-shot manner, where the model makes recommendations based on general pre-training [31, 57], or through fine-tuning the model on specific recommendation tasks [31]. While LLM zero-shot or few-shot methods might offer scrutability since they rely purely on natural text, they suffer from inconsistencies and are limited by confabulations

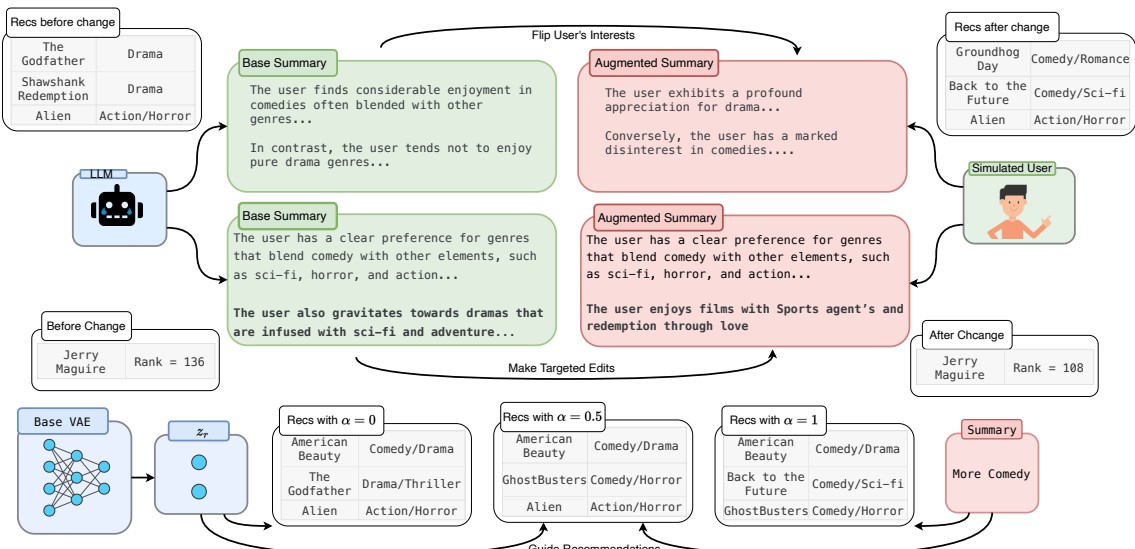

**Figure 3: Controllability experiment visualization: large-scope changes (top), fine-grained edits (middle), and guided recommendations (bottom). Red indicates edited summaries, green are base summaries and blue are models. Summaries and examples are paraphrased. App. Q includes more summaries, with examples in App. G for large-scope and App. H for fine-grained.**

and incomplete catalogue coverage [33]. We further validate this in Table 1, where we evaluate GPT-4-turbo's performance on strong generalization. On the other hand, fine-tuning can enhance the quality of recommendations by integrating new item-specific or user-specific tokens into the LLM's vocabulary [11, 12, 22]. However, it may compromise the model's ability to comprehensively navigate the full item catalogue due to the added tokens not appearing during general pre-training. While these approaches show that one can enhance performance using textual attributes, they do not focus on the development and evaluation of scrutable systems, which is the main focus of this work.

## 3 Methodology

We introduce TEARS, a method with user-interpretable and controllable representations. We begin by contrasting TEARS with standard latent-based CF methods. Then, we introduce the components of TEARS, starting with a prompting pipeline for summarizing the user preferences with an LLM. The summaries are then used to predict recommendations. To achieve this, we use two AE models: a text representation-based model, which transforms text summaries into recommendations, and a (black-box) VAE model. We propose aligning the space of the text representations model with the space of a standard recommendation VAE using optimal transport (OT). We find this alignment crucial for obtaining high-quality recommendations while preserving the controllability of the text user summaries (see App. M.1).

### 3.1 Motivation

Traditional collaborative-filtering-based recommender systems rely on a user's history to provide recommendations. A user wanting to obtain better recommendations, e.g. if current recommendations do not appear satisfying, faces a tedious process with unclear outcomes since they must interact with (e.g., consume or at least rate) items that better reflect their preferences with the hope of obtaining

better recommendations. This is even more impractical in domains where users' preferences evolve rapidly or if users have *short-term* preferences in a given context.

In contrast, TEARS allows users to adjust their recommendations by adapting or even deleting their summary and creating a new one more aligned with their (current) preferences. This process is immediate and transparent. It allows users to correct representational mistakes (e.g. add a missing genre they are interested in) or adapt them to better suit their evolving preferences and the current context. Additionally, we introduce an interpolation coefficient, $\alpha$, between a user's summary and black-box representations. Setting $\alpha = 1$ puts all the weight on text representations, while $\alpha = 0$ favors black-box representations. This gives users extra control over how their recommendations are influenced (details in Section 3.3).

*Background.* Autoencoder models have proven highly effective for collaborative-filtering recommender systems, consistently outperforming counterparts across various tasks [30, 44, 47]. With this in mind, we design TEARS to be compatible with existing VAE-based models and refer to the combination as TEARS VAE models. We study specific VAEs, and we denote their combinations using their names, e.g. TEARS RecVAE.

The auto-encoder framework involves representing the user-item feedback matrix $\mathbf{X} \in \mathbb{N}^{U \times I}$, where each entry represents a rating given by a user $u$ to an item $i$. Our focus is on predicting users' implicit preferences $\mathbf{Y} \in \{0, 1\}^{U \times I}$ (e.g. identifying items that a user has rated above a specified threshold $r$ as positive targets). These models prescribe learning an encoding function $Q : \mathbf{X} \rightarrow \mathbf{Z}$ to compress input data into a lower-dimensional latent space, followed by a decoding function $D : \mathbf{Z} \rightarrow \mathbf{Y}$ to map it to the target.

### 3.2 Summary Design

Creating scrutable summaries for controllable recommender systems presents unique difficulties. Manual summary creation is impractical due to scalability and inconsistency, while the quality

of earlier machine-generated summaries was low [8]. However, LLMs like the recent GPTs have significantly improved capabilities across natural language tasks [19], offering a tool for generating high-quality user summaries.

We propose designing user summaries by leveraging LLMs. While these generative models provide an efficient way to obtain summaries, ensuring their quality and consistency is non-trivial.

We believe each summary should contain enough information to be decodable into good recommendations but short enough to be easy to understand and to control by users. In that sense, it should describe the user's preferences sufficiently and uniquely. We note that these design choices may also vary by domain, and in this work, we focus on the movie and book recommendation domains.

Given the above criteria, we identify preference attributes that a user may wish to edit and that are essential in providing good recommendations. For each attribute, we also pinpoint relevant prompting information:

- **Inferred Preferences**: What users like and dislike, prompted with user ratings.
- **High-Level Attributes**: Preferences for genres, prompted with item metadata.
- **Fine-Grained Details**: Specific plots or themes, prompted with the title.

Recent LLMs encode significant knowledge about movies and books [18, 20, 53]. We believe they should be able to encode appropriate item information conditioned on titles and genres alone. To verify this, we conduct preliminary experiments using GPT-3.5, GPT-4-turbo, and GPT-4 via the OpenAI API, finding that GPT-3.5 generated poor summaries, while there was no significant difference between GPT-4 and GPT-4-turbo. We select GPT-4-turbo[2] to generate summaries and refer to it as GPT for the remainder of the text. Additionally, to enhance the reproducibility of our work, we also generate summaries using LLaMA 3.1-405b[3] [7].

While LLMs have shown impressive capabilities in text summarization [8], we find that free-style prompting without adherence to a specific structure can make summaries generic and vary in quality. On the other hand, LLMs have excelled in instruction-based tasks [38]. With this in mind, we design a prompt asking for summaries to include the desirable characteristics mentioned to enforce consistency and quality. Consistent summaries will also likely help train a decoder and obtain high-quality recommendations. Our resulting prompting strategy is in Figure 14. We explicitly direct the model to avoid stating titles and rating information to prevent over-reliance on such details and encourage summaries to be more expressive. By design, LLM's responses are non-deterministic. In early experiments, we find that summary generation can vary, with two output summaries being significantly different for the same user. We observe higher variability in users with longer histories, leading us to limit the number of items used for each summary (to a maximum of 50 items in our studies), we further explore this effect in App N. Note that the number of items used for each summary, $m_u$, is user-dependent since some users have shorter histories that this max. threshold. Finally, our prompt also contains the expected length of the summaries, which we set to 200 words, as we

find it short enough to not incur heavy cognitive loads, but can be detailed enough. For future work, we leave such explorations, including whether the summary length should vary by user.

## 3.3 TEARS

In this section, we define TEARS and its training process, which is depicted in Figure 1. With user summaries $\mathbf{S}$ and feedback data $\mathbf{X}$, our goal is to obtain a pair of encoding functions $Q_s : \mathbf{S} \rightarrow Z_s$ and $Q_r : \mathbf{X} \rightarrow Z_r$ which we can constrain to map the representations $z_s^u$ and $z_r^u$ to a common space. We obtain $Q_r$ from a backbone VAE for recommendations. After that, we aim to decode a convex combination of the representations $z_c^u = \alpha z_s^u + (1 - \alpha) z_r^u$ onto recommendations using a shared decoder $D : Z_c \rightarrow \mathbf{Y}$. When $\alpha = 1$, the recommendations are generated solely using the summary embeddings; this means the downstream recommendations are controllable through simple text edits. On the other hand, when $\alpha = 0$, the recommendations are based purely on the backbone VAE and only leverage the user feedback data. Other $\alpha$ values lead to a combination of these, making it such that a user can guide their recommendations through text edits but still use their historical data, which may be richer in information, making the changes less drastic but more personalized. Overall, our training objective is composed of three components, which are detailed below.

*Alignment through Optimal Transport.* While the shared decoder architecture should incentivize both the text ($Z_s$) and black-box embeddings ($Z_r$) to be naturally aligned, in practice we find that training without additional constraints is not enough (see App. M.1). Rather, we align these embeddings using optimal transport techniques which measure the cost of shifting the mass from one probability measure to another [3]. This is achieved by calculating a cost function that reflects the underlying geometry of the distributions, known as the Wasserstein distance. Unlike other distance metrics such as KL-divergence, the Wasserstein distance is symmetric, making it particularly suitable as an optimization target for aligning two distributions. Computing this distance with Gaussian distributions has a closed-form solution [25]. To make use of these properties, we use encoders $Q_r$ and $Q_s$ that map inputs onto Gaussian-distributed latent encodings, as is traditional for VAEs [24, 30, 45], $Z_r \sim N(\mu_r, \sigma_r \mathbf{I})$ and $Z_s \sim N(\mu_s, \sigma_s \mathbf{I})$. This parameterization allows for direct computation of the minimal transportation cost between Gaussian distributions to align the two embeddings:

$$\mathcal{L}_{OT} = ||\mu_s - \mu_r||_2^2 + \text{Tr}\{\Sigma_s + \Sigma_r - 2(\Sigma_r^{\frac{1}{2}} \Sigma_S \Sigma_r^{\frac{1}{2}})^{\frac{1}{2}}\}. \quad (1)$$

Other optimal transport techniques, like Sinkhorn's algorithm [6], are applicable to non-Gaussian distributions and we reserve these methods for future exploration. We find in practice, this objective greatly enhances the system's controllability (see App. M.5).

*Objective for Recommendation* For $Q_s$, we use a T5-base model [41], which we fine-tune using low-rank adaptors [21], to obtain an embedding of the text summaries and train an MLP head to obtain $\mu_s, \sigma_s$, we then use the reparametrization trick to obtain $Z_s$:

$$\mu_s, \sigma_s = \text{MLP}\big(\text{T5-Encoder}(S)\big), \quad (2)$$

$$Z_s = \mu_s + \sigma_s \circ \epsilon, \quad \epsilon \sim N(0, \mathbf{I}). \quad (3)$$

Thanks to the OT alignment, $Z_r$, $Z_s$ and $Z_c$ share a common space and thus a shared decoder, $D$ alongside the softmax function $\Psi$ can

---

[2]We use gpt-4-1106-preview through the OpenAI API https://platform.openai.com/
[3]We obtain summaries using the help of https://www.llama-api.com/

be used to produce a distribution over items for each user from each latent representation $(Z_s, Z_r)$ and their combination $(Z_c)$:

$$\hat{Y}_c = \Psi(D(Z_c)), \ \hat{Y}_r = \Psi(D(Z_r)), \ \hat{Y}_s = \Psi(D(Z_s)). \quad (4)$$

We use these distributions to optimize the multinomial likelihood of each representation. During training, we fix $\alpha = 0.5$ to optimize for performance on the merged representations but note that $\alpha$ can be changed at any time during inference. The model learns using the binary cross-entropy of trained-autoencoder (r), TEARS (s), and their combination:

$$\mathcal{L}_R = \sum_{k \in \{c,s,r\}} \sum_{i \in I, u \in U} y_{ui} \log(\hat{y}_{ui,k}). \quad (5)$$

*Constraint of Gaussian Priors* Additionally, we impose a standard Gaussian prior $P(Z) \sim N(\mathbf{0}, \mathbf{I})$ on $Z_s$ which has been shown to help improve performance [30]. Enforcing this constraint can be expressed as optimizing the KL-divergence between that prior and its inferred value:

$$\mathcal{L}_{KL} = DKL(Q_s(Z \mid S) || P(Z))$$

Our overall training objective is a weighted sum of the above three objectives, formulated as below:

$$\mathcal{L} = \mathcal{L}_R + \lambda_1 \mathcal{L}_{OT} + \lambda_2 \mathcal{L}_{KL}, \quad (6)$$

where $\lambda_1$ and $\lambda_2$ are weighing parameters for their respective losses. In practice, we initialize $D$ with the base model's decoder, update its weights while training, and freeze $Q_r$'s weights (see App. M.3).

## 3.4 Genre-Based Model (GERS)

In addition to summary-based TEARS VAE models, we instantiate GERS VAE models, which help evaluate whether TEARS summaries contain information beyond genres. The only difference between TEARS VAE and GERS VAE is that the text summaries $S$ and user representation $z_s^u$ are replaced with a genre vector $G$ and a genre-based representation $z_g^u$, defined as the normalized count vector of the genres linked to the items a user has interacted with positively. Specifically, we define this representation as:

$$z_{g,\rho}^u = \frac{C_u(\rho)}{\sum_{\rho' \in \mathcal{G}} C_u(\rho')} \quad (7)$$

where $\mathcal{G}$ represents the set of all genres, and $C_u(\rho)$ corresponds to the count of items the user has interacted with in genre $\rho$, such that $z_g^u \in [0,1]^{|\mathcal{G}|}$. This modeling framework is similar to those used in keyword/tag-based systems, as each genre entry in $z_g^u$ can be scrutinized by the user making it so they can choose how much of that genre they want to be weighted in their recommendations (see App. J). We note, that while this modeling framework can be useful for some applications, it is limited in transparency as user representations are simple statistics. Moreover, users cannot edit fine-grained details of their interests such as plots or themes they may enjoy, limiting the controllability of this system.

## 4 Datasets

We conduct experiments on subsets of the MovieLens-1M (ML-1M),[4] Netflix,[5] and Goodbooks[6] datasets. As is common in other

----

[4] https://grouplens.org/datasets/movielens/1m/
[5] https://www.kaggle.com/datasets/netflix-inc/netflix-prize-data
[6] https://github.com/zygmuntz/goodbooks-10k

studies, we filter out cold-start items for all datasets [16, 54]. In addition, due to the high cost of using LLM APIs, we use a subset of users with enough ratings for each dataset to provide a comprehensive summary. For the Netflix and Goodbooks datasets, we filter out users with less than 100 interactions and items with less than twenty. For the smaller ML-1M dataset, we only filter out users with less than twenty interactions and items with less than 5. After filtering, we have 6,037 users and 2,745 items for ML-1M, 9,978 users and 3,081 items for Netflix, and 9,980 users and 8,093 items for Goodbooks. Descriptive statistics such as sparsity, average ratings, and number of genres per dataset are reported in App B.

We construct $\mathbf{Y}$ using $\mathbf{X}$, with $r = 4$, that is, we train the model to predict implicit feedback where the rating is positive ($y_{ui} = 1$) if the item is rated four and above and a negative ($y_{ui} = 0$) otherwise [34]. We evaluate under a strong generalization setting where we reserve 500 users for the validation and testing splits (250 each) for ML-1M and 2,000 (1,000 each) for Netflix and Goodbooks. We make all summaries with GPT and LLaMA using the prompt in Figure 14. We use up to 50 of the oldest ratings to construct the summaries, while the remaining, more recent ones are used for evaluation. For users that rate less than fifty items, we retain the most recent two for evaluation and generate the summary with the remaining.

## 5 Assessment of User Summaries

We begin by assessing the scrutability and uniqueness of user summaries, using descriptive statistics on their length alongside standard NLP metrics. Thereafter, we use recommendation performance as a proxy for quality, as accurate, information-rich summaries should yield better recommendations.

## 5.1 User Summary Properties

We analyze the average summary length to inspect if summaries can be appropriately scrutinized. For GPT, summaries average 168.67±3.62 words across all datasets, compared to 179.60±2.20 for LLaMA. These lengths suggest that the summaries are concise enough to be easily editable while still comprehensive enough to convey detailed user information, as reflected in their positive impact on recommendation performance (see §5.2). To assess uniqueness, we use pairwise edit distance and BLEU scores [39], the latter measuring n-gram overlap between two texts (we use 4-grams). The average edit distance for GPT summaries is 160.17±0.05 words, while LLaMA summaries average 157.05±1.18. These scores are high when compared to the average summary length, indicating distinct summaries across users. Similarly, BLEU scores are low, with GPT summaries averaging 0.08±0.02 and LLaMA summaries 0.19±0.01, which suggest minimal n-gram overlap, so diverse phrasing. Further details and statistics are in App. A and examples in App. Q.

## 5.2 Recommendation Quality

We assess the quality of information within user summaries using recommendation performance as a proxy. For this, we benchmark popular AE-based methods, GPT, and two TEARS and GERS variants, using recall@k and NDCG@k established top-$k$ metrics.

*Models.* We evaluate several AE-based models, including Multi-VAE [30], RecVAE [45], MacridVAE [32], EASE [47], and Multi-DAE [30]. To ensure fairness in comparison, we use the same ratings used to generate the user summaries as input for these models. Unlike

Table 1: Comparison of model performance across the ML-1M, Netflix, and Goodbooks datasets on NDCG and recall at $k \in \{20, 50\}$. We label LLaMA models with ∞ and GPT models with ◉. Each model is evaluated using five different seeds. We report both mean values and standard deviations. Best results are denoted in bold, and a * indicates statistical significance ($p < 0.05$) in a two-way t-test between the TEARS VAE model and its respective VAE (above in grey).

| Model | ML-1M | | | | Netflix | | | | Goodbooks | | | |
|---|---|---|---|---|---|---|---|---|---|---|---|---|
| | Recall@20 | NDCG@20 | Recall@50 | NDCG@50 | Recall@20 | NDCG@20 | Recall@50 | NDCG@50 | Recall@20 | NDCG@20 | Recall@50 | NDCG@50 |
| GPT-4-turbo | 0.031 | 0.033 | 0.048 | 0.0390 | 0.054 | 0.067 | 0.065 | 0.040 | 0.015 | 0.012 | 0.013 | 0.011 |
| EASE [47] | 0.295 | 0.277 | 0.320 | 0.270 | 0.496 | 0.518 | 0.441 | 0.466 | 0.173 | 0.180 | 0.193 | 0.182 |
| Multi-DAE [30] | $0.290_{\pm 0.002}$ | $0.254_{\pm 0.001}$ | $0.363_{\pm 0.004}$ | $0.266_{\pm 0.000}$ | $0.507_{\pm 0.001}$ | $0.532_{\pm 0.001}$ | $0.450_{\pm 0.000}$ | $0.476_{\pm 0.001}$ | $0.151_{\pm 0.002}$ | $0.155_{\pm 0.002}$ | $0.173_{\pm 0.001}$ | $0.160_{\pm 0.001}$ |
| GERS Base | $0.276_{\pm 0.003}$ | $0.246_{\pm 0.001}$ | $0.320_{\pm 0.004}$ | $0.248_{\pm 0.000}$ | $0.471_{\pm 0.001}$ | $0.497_{\pm 0.001}$ | $0.413_{\pm 0.001}$ | $0.440_{\pm 0.001}$ | $0.153_{\pm 0.001}$ | $0.161_{\pm 0.001}$ | $0.167_{\pm 0.001}$ | $0.161_{\pm 0.001}$ |
| ◉ TEARS Base | $0.267_{\pm 0.004}$ | $0.253_{\pm 0.002}$ | $0.302_{\pm 0.004}$ | $0.250_{\pm 0.005}$ | $0.465_{\pm 0.004}$ | $0.491_{\pm 0.004}$ | $0.413_{\pm 0.003}$ | $0.439_{\pm 0.004}$ | $0.145_{\pm 0.003}$ | $0.153_{\pm 0.002}$ | $0.158_{\pm 0.002}$ | $0.153_{\pm 0.002}$ |
| ∞ TEARS Base | $0.259_{\pm 0.010}$ | $0.249_{\pm 0.010}$ | $0.292_{\pm 0.015}$ | $0.245_{\pm 0.010}$ | $0.452_{\pm 0.002}$ | $0.479_{\pm 0.002}$ | $0.397_{\pm 0.001}$ | $0.424_{\pm 0.002}$ | $0.143_{\pm 0.002}$ | $0.151_{\pm 0.003}$ | $0.156_{\pm 0.002}$ | $0.151_{\pm 0.002}$ |
| ∞ TEARS RecVAE $_{\alpha=1}$ | $0.307_{\pm 0.006}$ | $0.272_{\pm 0.005}$ | $0.351_{\pm 0.007}$ | $0.276_{\pm 0.005}$ | $0.483_{\pm 0.002}$ | $0.509_{\pm 0.001}$ | $0.428_{\pm 0.002}$ | $0.455_{\pm 0.001}$ | $0.150_{\pm 0.002}$ | $0.160_{\pm 0.003}$ | $0.163_{\pm 0.001}$ | $0.159_{\pm 0.001}$ |
| Multi-VAE [30] | $0.295_{\pm 0.002}$ | $0.261_{\pm 0.001}$ | $0.357_{\pm 0.002}^{*}$ | $0.270_{\pm 0.001}$ | $0.507_{\pm 0.001}$ | $0.532_{\pm 0.001}$ | $0.450_{\pm 0.000}$ | $0.476_{\pm 0.001}$ | $0.159_{\pm 0.001}$ | $0.163_{\pm 0.001}$ | $0.186_{\pm 0.001}$ | $0.170_{\pm 0.001}$ |
| ◉ TEARS Multi-VAE $_{\alpha*}$ | $0.295_{\pm 0.003}^{*}$ | $0.267_{\pm 0.002}^{*}$ | $0.344_{\pm 0.010}$ | $0.272_{\pm 0.003}$ | $0.512_{\pm 0.001}^{*}$ | $0.538_{\pm 0.001}^{*}$ | $0.451_{\pm 0.000}^{*}$ | $0.480_{\pm 0.000}^{*}$ | $0.171_{\pm 0.002}^{*}$ | $0.178_{\pm 0.002}^{*}$ | $0.187_{\pm 0.003}$ | $0.178_{\pm 0.002}^{*}$ |
| ∞ TEARS Multi-VAE $_{\alpha*}$ | $0.306_{\pm 0.003}^{*}$ | $0.276_{\pm 0.003}^{*}$ | $0.347_{\pm 0.007}$ | $0.278_{\pm 0.003}^{*}$ | $0.510_{\pm 0.001}^{*}$ | $0.536_{\pm 0.001}^{*}$ | $0.450_{\pm 0.001}$ | $0.479_{\pm 0.001}^{*}$ | $0.169_{\pm 0.002}^{*}$ | $0.174_{\pm 0.002}^{*}$ | $0.187_{\pm 0.003}$ | $0.176_{\pm 0.002}^{*}$ |
| MacridVAE [32] | $0.301_{\pm 0.007}$ | $0.260_{\pm 0.006}$ | $0.370_{\pm 0.002}$ | $0.276_{\pm 0.005}$ | $0.505_{\pm 0.003}$ | $0.529_{\pm 0.003}$ | $0.450_{\pm 0.002}$ | $0.476_{\pm 0.001}$ | $0.168_{\pm 0.001}$ | $0.170_{\pm 0.001}$ | $\mathbf{0.196_{\pm 0.001}}$ | $0.178_{\pm 0.001}$ |
| ◉ TEARS MacridVAE $_{\alpha*}$ | $\mathbf{0.323_{\pm 0.004}^{*}}$ | $0.280_{\pm 0.004}^{*}$ | $\mathbf{0.381_{\pm 0.006}^{*}}$ | $\mathbf{0.291_{\pm 0.003}^{*}}$ | $0.511_{\pm 0.001}^{*}$ | $0.535_{\pm 0.002}^{*}$ | $0.454_{\pm 0.002}^{*}$ | $0.480_{\pm 0.002}^{*}$ | $0.171_{\pm 0.002}^{*}$ | $0.175_{\pm 0.002}^{*}$ | $0.195_{\pm 0.002}$ | $0.180_{\pm 0.001}^{*}$ |
| ∞ TEARS MacridVAE $_{\alpha*}$ | $0.319_{\pm 0.004}^{*}$ | $0.280_{\pm 0.002}^{*}$ | $0.376_{\pm 0.003}^{*}$ | $0.289_{\pm 0.001}^{*}$ | $0.510_{\pm 0.001}^{*}$ | $0.536_{\pm 0.001}^{*}$ | $0.450_{\pm 0.001}$ | $0.479_{\pm 0.001}^{*}$ | $0.169_{\pm 0.001}^{*}$ | $0.173_{\pm 0.001}^{*}$ | $0.194_{\pm 0.002}$ | $0.179_{\pm 0.001}$ |
| RecVAE [45] | $0.300_{\pm 0.005}$ | $0.264_{\pm 0.003}$ | $0.360_{\pm 0.003}$ | $0.274_{\pm 0.003}$ | $0.515_{\pm 0.003}$ | $0.540_{\pm 0.003}$ | $0.455_{\pm 0.002}$ | $0.482_{\pm 0.002}$ | $0.171_{\pm 0.001}$ | $0.176_{\pm 0.001}$ | $0.191_{\pm 0.002}$ | $0.179_{\pm 0.001}$ |
| GERS RecVAE $_{\alpha*}$ | $0.304_{\pm 0.003}^{*}$ | $0.266_{\pm 0.003}^{*}$ | $0.366_{\pm 0.004}^{*}$ | $0.279_{\pm 0.002}^{*}$ | $0.517_{\pm 0.001}^{*}$ | $0.542_{\pm 0.001}^{*}$ | $\mathbf{0.458_{\pm 0.001}^{*}}$ | $\mathbf{0.485_{\pm 0.002}^{*}}$ | $0.170_{\pm 0.001}$ | $0.176_{\pm 0.001}$ | $0.192_{\pm 0.001}$ | $0.180_{\pm 0.001}$ |
| ◉ TEARS RecVAE $_{\alpha*}$ | $0.307_{\pm 0.002}^{*}$ | $0.273_{\pm 0.002}^{*}$ | $0.374_{\pm 0.002}^{*}$ | $0.285_{\pm 0.001}^{*}$ | $0.517_{\pm 0.001}^{*}$ | $0.543_{\pm 0.000}^{*}$ | $0.457_{\pm 0.001}^{*}$ | $0.485_{\pm 0.001}^{*}$ | $\mathbf{0.175_{\pm 0.002}^{*}}$ | $\mathbf{0.181_{\pm 0.002}^{*}}$ | $0.193_{\pm 0.000}^{*}$ | $\mathbf{0.183_{\pm 0.001}^{*}}$ |
| ∞ TEARS RecVAE $_{\alpha*}$ | $0.319_{\pm 0.005}^{*}$ | $\mathbf{0.282_{\pm 0.005}^{*}}$ | $0.363_{\pm 0.003}^{*}$ | $0.287_{\pm 0.002}^{*}$ | $\mathbf{0.518_{\pm 0.001}}$ | $\mathbf{0.544_{\pm 0.001}^{*}}$ | $0.457_{\pm 0.001}^{*}$ | $0.485_{\pm 0.001}^{*}$ | $0.173_{\pm 0.001}^{*}$ | $0.179_{\pm 0.001}^{*}$ | $0.191_{\pm 0.002}$ | $0.181_{\pm 0.000}^{*}$ |

the typical practice of binarizing inputs, we use the original ratings, as this better aligns with the user summaries and leads to improved performance in AEs. Additionally, we benchmark TEARS VAEs with backbones built from Multi-VAE, RecVAE, and MacridVAE. For GERS, we only train it using RecVAE as the backbone VAE, as we found it to be the best-performing AE. Additionally, we evaluate versions of TEARS and GERS without a backbone VAE—referred to as TEARS Base and GERS Base. TEARS Base most closely resembles the framework visualized in Figure 1 [40]. We also evaluate GPT under strong generalization using few-shot prompting to compare against a fully LLM-based solution (details are in App. D).

*Discussion.* Table 1 presents the results, averaged over five seeds evaluated with the top $k = 20, 50$ items. For TEARS and GERS VAE models, we report test set metrics using $\alpha*$, being the optimal value determined using the validation set (see App. K for $\alpha$ values). Additionally, we compare against Base models by including results for the LLaMA-based TEARS RecVAE using $\alpha = 1$, being the most performant when only using summary embeddings (see App L).

We find TEARS VAE models consistently outperform their backbone AEs across all but a single benchmark. This increase in performance is most prevalent in ML-1M where users are generally less active as well as in a sparser dataset like Goodbooks (see App B). This suggests summaries have useful information not found in the black-box embeddings contributing to performance. We find that TEARS VAEs can consistently improve upon their backbone VAEs. This is even the case with RecVAE, the best-performing AE, where TEARS RecVAE outperforms it on all datasets regardless of the backbone LLM. Notably, on ML-1M, TEARS RecVAE can outperform some black-box models using an $\alpha = 1$. It is also encouraging that TEARS VAE models can offer higher performance while maintaining scrutability (see § 6 for these results). In contrast, TEARS Base does considerably worse than all AEs, showcasing the importance of TEARS VAEs' hybrid setup. GPT performs poorly in this context, reinforcing the need for model adaptation to this task.

Meanwhile, GERS Base consistently outperforms TEARS Base, suggesting that simple genre statistics can be more effective than summaries for initial recommendations. However, TEARS RecVAE

outperforms GERS RecVAE on the ML-1M and Goodbooks datasets, indicating that user summaries provide valuable information beyond black-box or genre representations. On Netflix, however, TEARS RecVAE and GERS RecVAE perform similarly, implying that the performance gains are likely explained by genre information. To improve this, future work could explore more dataset-specific prompts, but it is noteworthy that even in cases where summaries have a limited impact, they still offer slight performance improvements and, crucially, do not degrade performance. Furthermore, TEARS RecVAE with $\alpha = 1$ consistently outperforms TEARS Base, demonstrating that aligning black-box and summary embeddings enhances performance when using only user summaries.

Overall, TEARS VAE models show that it is possible to significantly enhance scrutability without sacrificing—and even improving—performance compared to both AE and genre-based models. We provide additional analysis over components of TEARS, such as ablation studies in App. M, analysis on user activity levels in App O, and evaluate GPT-based models using LLaMA summaries and vice versa in App. P.

## 6 Controllability Through Text Edits

We now study the controllability of user text representations, which is the ability of users to edit and readjust their representation to (better) align the system's recommendations with their preferences. Controllability is one of the main advantages of text representations compared to latent representations. We analyze GPT-based models and summaries in the main text and provide all results for LLaMA-based models in App. F. As these changes are specific to textual summaries, we do not assess the controllability of the genre-based models in the following experiments but provide analyses in App. J.

Given the lack of evaluation metrics for scrutable recommendations, we create three tasks to evaluate user controllability. Each task corresponds to a scenario that would lead users to update their text summaries. The resulting tasks are easily benchmarked across methods. First, *large-scope changes*, for example, to correct significant inaccuracies in a user profile. Second, finer or *small-scope* changes aim to readjust minor discrepancies in a user summary.

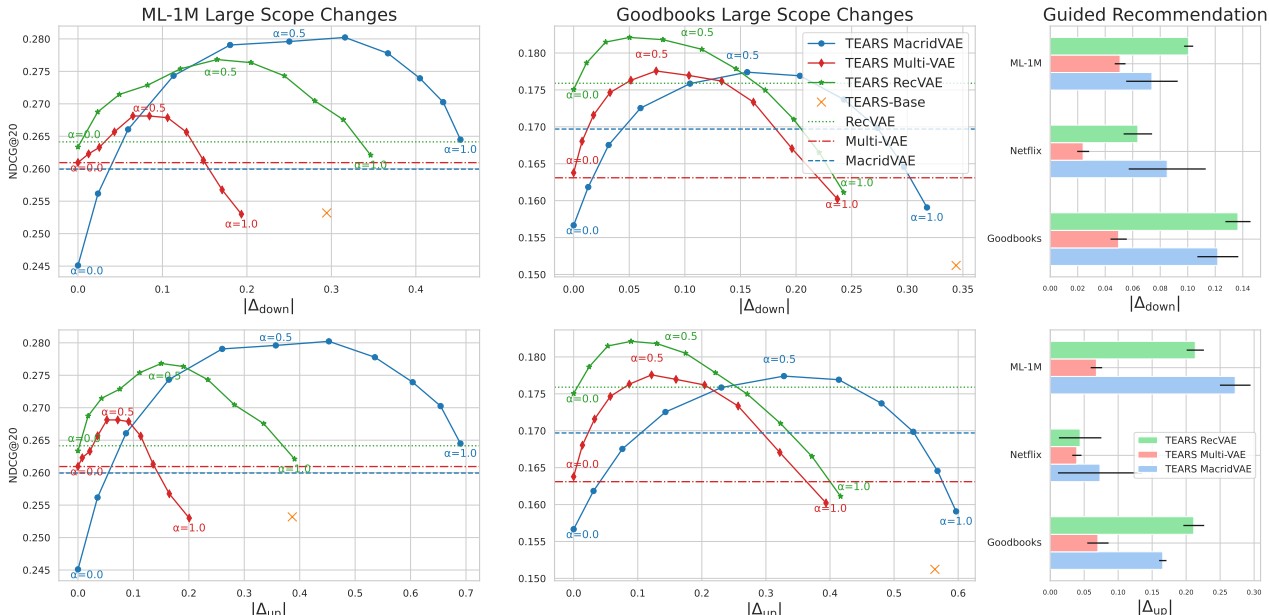

**Figure 4: Tradeoff between recommendation performance (y-axis) and large scope controllability (x-axis) for ML-1M (left) and Goodbooks (middle) using GPT-generated summaries (see Appendix F for LLaMA results). Netflix data is shown in Figure 8. The x-axis represents $|\Delta_{\text{up/down}}|$ as $\alpha$ increases, reflecting its impact on NDCG@20. Notably, most TEARS VAE models outperform TEARS BASE in recommendation performance at $\alpha = 1$, with some also achieving superior controllability. The bar plots (right) illustrate guided recommendation outcomes, where all models successfully guide black-box embeddings in the intended direction. Results are averaged across five seeds, with standard deviations detailed in Appendix K.**

Other changes can be seen as interpolating between these two cases where a large change is simply many aggregate small edits. Third, we test the ability of summaries to guide personalized recommendations. This tests a different type of user interaction where the summary is used as an instruction (e.g. in a particular context). This evaluates a model's capacity to interpolate between historical behavior and a context. All experiments are visualized in Figure 3.

## 6.1 Evaluating Large Scope Changes

We first evaluate how well TEARS can react to a large change in a user's interest. To simulate such a change, we prompt GPT to "flip" a user's interest. We do so by first prompting GPT to identify a user's most and least favored genre. Using these genres, we prompt GPT to make the user's favorite genre into its least favorite and vice-versa, effectively inducing a large shift in the user's preference. We find that GPT can appropriately induce a shift in user preference. We provide an example and the full prompting strategy in App. G. To evaluate TEARS' effectiveness at modeling such changes, we design the genre-wise Discounted Cumulative Gain at $k$ ($\text{DCG}_g@k$), which measures how favored a genre $\rho$ is in the user's top-$k$ rankings. Items from a newly-favored genre should rank higher than in the original ranking, and the difference in $\text{DCG}_g$ captures this shift.

Below, we define $\text{DCG}_g@k$, where $\omega(i)$ maps the $i$-th item to its corresponding set of genres (items can have multiple genres):

$$\text{DCG}_g@k(\rho) = \sum_{i=1}^{k} \frac{\mathbf{I}(\rho \in \omega(i))}{\log_2(i+1)}. \tag{8}$$

We normalize $\text{DCG}_g@k$ using the Idealized Discounted Gains (IDCG) to obtain the genre-wise NDCG ($\text{NDCG}_g@k$). To assess the effectiveness of the changes, we measure the $\Delta@k$ change in $\text{NDCG}_g@k$ between the original (denoted with a superscript O) and augmented summary (denoted with superscript A):

$$\Delta@k(\rho) = \text{NDCG}_g^O@k(\rho) - \text{NDCG}_g^A@k(\rho). \tag{9}$$

We evaluate each summary using two metrics: $|\Delta_{\text{up}}@k|$, which assesses TEARS' ability to elevate the rankings of the initially least favored genre ($\rho$ = least favorite), and $|\Delta_{\text{down}}@k|$, which gauges its proficiency in lowering the rankings of the initially favored genre ($\rho$ = favorite). Additionally, we explore how the parameter $\alpha$ influences controllability, highlighting the trade-off between recommendation performance and controllability. We prompt GPT to obtain the altered summaries for all test users and use those to examine shifts in genre preferences. Figure 4 illustrates the trade-off between recommendation performance (NDCG@20) calculated using the **original summaries** compared to controllability ($|\Delta_{\text{up}}@20|$ & $|\Delta_{\text{down}}@20|$) as $\alpha$ increases, averaged over five seeds, for ML-1M and Goodbooks (Netflix in App F.2). We find that controllability increases as $\alpha$ does, with all TEARS variants having meaningful levels of controllability at higher $\alpha$ values across all datasets, regardless of backbone LLM (see App. F). Remarkably, TEARS VAEs maintain satisfactory controllability even at reduced values of $\alpha$. As $\alpha$ increases, we find that NDCG@20 peaks at some intermediate value visually resembling an inverse-U shape, likely due to $\mathcal{L}_R$ (Eq. 5) optimizing for performance over various values of $\alpha$. We find TEARS MacridVAE

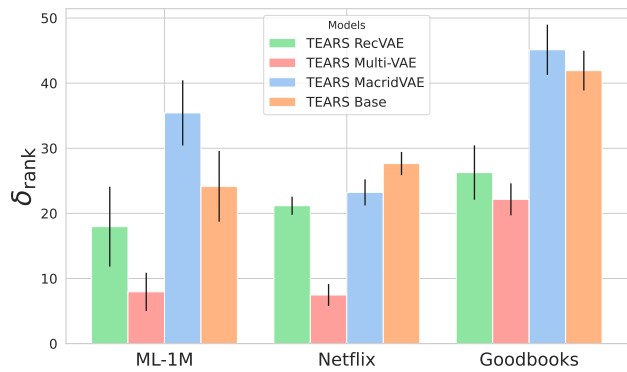

**Figure 5: $\delta_{\mathrm{rank}}$ after fine-grained changes. Target items can gain tens of positions from small edits in user summaries. Here error bars represent the standard error $\frac{\sigma}{\sqrt{n}}$.**

to be the best model, consistently outperforming TEARS Base in controllability and AEs in NDCG@20 across multiple values of $\alpha$.

## 6.2 Fine-Grained Changes

We now evaluate smaller edits to user summaries by simulating a task where a user wants to increase the rank of a single target item by making small edits to their summary. While we could do such a simulation by putting in the item's name, actors, or description, we are not interested in such use cases which other systems, such as search engines, are better suited for. Rather, we simulate summary changes alluding to higher-level characteristics such as plot points or themes, that could be linked to many items. To achieve this, we sample an item from the evaluation set, ranked between positions 100 and 500. This range was chosen because it suggests the summary may omit certain item attributes that could improve its rank. We make sure the item is within this range for all models within all values of $\alpha$. With these sampled items, we prompt GPT with two tasks: first, to "summarize the item in 5 words while only referring to plot points/themes," and then to replace an existing sentence in the summary with one including these plot points and themes. Using this, we measure the difference in rank $\delta_{\mathrm{rank}}$ between the original and the rank after the change. Given the variability of LLMs, we rerun the procedure three times and report the median $\delta_{\mathrm{rank}}$ as an estimator of whether users can have positive interactions with the system more often than negative ones. We filter out users who do not have an item that satisfies such criteria, leaving 150 users for ML-1M, 886 for Netflix, and 690 for Goodbooks.

Figure 5 reports the rank differences between the augmented and original summaries when $\alpha = 1$. We observe that for all datasets, all models can increase the rank of the target item by tens of positions ($\delta_{\mathrm{rank}} > 0$) with minor changes to the summary. Moreover, we find TEARS Base and TEARS MacridVAE are the best performing across datasets, with Multi-VAE being the weakest, a finding consistent with the results of §6.1. Across all datasets and models, we can consistently move a target item, even with small changes within summaries. In-App. H we provide a complete overview over varying values of $\alpha$, examples, and other implementation details.

## 6.3 Guided Recommendations

We design a task to assess if users can use short instructions as their summaries to guide/obtain contextual recommendations. This

change is particularly interesting for TEARS VAE models as they can use an $\alpha \neq 1$ to guide their black-box embeddings with a small amount of text, for instance, a user might request that the system include "more action movies" in its current recommendations. To evaluate if TEARS can effectively deliver such targeted recommendations, we design an experiment where we measure $\Delta_{\mathrm{up/down}}$ where $NDCG@20_g^O$ is measured using the black-box embeddings (i.e. $Z_{\alpha=0}$) and $NDCG@20_g^A$ is calculated using $Z_{\alpha=0.5}$, where the summary is a simple guidance prompt indicating "More {genre} {item_type}." Here using $Z_{\alpha=0.5}$ suggests that adding a genre preference should yield personalized recommendations favouring that genre, as the representation has to adhere to the base black-box representations. Similarly, we aim to simulate moving target genres down using phrases like, "Less {genre} {item_type}.", which initial analysis showed is a much harder task, as merely mentioning a genre was sufficient to elevate related items when no other summary information was provided. To address this, we adjust the model by subtracting negative text representations from black-box representations, $Z_{-\alpha=0.5} = \frac{Z_r - Z_s}{2}$. For this experiment, $NDCG@20_g^A$ is calculated using the rankings generated by $Z_{-\alpha=0.5}$. In practice, this procedure could be implemented using a sentiment classifier[55] to identify whether the system should use $Z_{-\alpha=0.5}$ or $Z_{\alpha=0.5}$. We evaluate these changes for all test users using the ten genres with the most corresponding items in each dataset.

The rightmost plots of Figure 4 display the results for the guided recommendation experiment over the three datasets. We note that the $|\Delta|$ changes (i.e the increase in the contextualized genre) we see are expected to be smaller as we use an $\alpha = 0.5$ for this setting. Nonetheless, we see that even with summaries composed of simple phrases, we can guide recommendations in the desired direction. Moreover, these changes in recommendations are more personalized as they are a combination of the base black-box embeddings and the altered summary. This procedure, and the latents of the black-box embeddings, are further visualized in Appendix I.

## 7 Conclusion and Discussion

We present TEARS, a method for constructing controllable recommender systems using natural-text user representations. By aligning user-summary and black-box embeddings through OT techniques, we demonstrate that the system provides higher-quality recommendations and is controllable. For the latter, we identify three types of changes users can make to their summaries to impact their downstream recommendations: *large-scope changes, fine-grained edits* and *guided recommendations*. To evaluate the controllability of TEARS under these changes, we design evaluation metrics and simulated tasks. The results show that TEARS models are controllable in each task. These tasks are reproducible and can be benchmarked in future work. While creating a user interface for TEARS and designing experiments to enable user studies is out of scope, it presents opportunities for future work. Overall, this work shows that scrutability can contribute to performance and leads to a new class of recommender systems that are more transparent and controllable. This work also opens new ways for users to interact with recommender systems, which we hope future work will develop.

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

## A  Summary Characteristics

Table 2 displays various qualities of the generated summaries. Overall we find that the average summary length is under 200 words, demonstrating the conciseness of the generated content. However, despite GPT's general adherence to instructions, we observe that the variance in summary lengths is greater than ideal, with some summaries exceeding the expected 200-word limit. For future work, we recommend more refined prompt engineering and potentially fine-tuning the LLM for summary construction to enhance consistency in summary length and adherence to target constraints.

Table 2: Summary length statistics. We compute average statistics for each dataset. We find GPT on average adheres to instructions, but has high variance in its output. We observe that the BLEU scores are quite low, and edit distances are comparable to the average summary length. Both these findings suggest the summaries are distinct between users, enhancing personalization.

|  | ML-1M | | Netflix | | Goodbooks | |
|---|---|---|---|---|---|---|
|  | GPT-4-preview | LLaMA 3.1 | GPT-4-preview | LLaMA 3.1 | GPT-4-preview | LLaMA 3.1 |
| Max Length | 268 | 266 | 268 | 257 | 246 | 257 |
| Minimum Length | 83 | 70 | 43 | 71 | 79 | 71 |
| 90th Percentile Length | 205 | 219 | 203 | 220 | 197 | 220 |
| 10th Percentile Length | 141 | 136 | 140 | 140 | 125 | 140 |
| Average Length | 171.27 ±26.47 | 176.49 ±31.96 | 170.20 ±26.38 | 181.15 ±30.62 | 164.52 ±27.60 | 181.15 ±30.62 |
| Edit Distances | 160.25 ±23.06 | 156.21 ±18.58 | 172.45 ±21.18 | 156.21 ±18.58 | 160.13 ±18.89 | 156.21 ±18.58 |
| BLEU Scores | 0.07 ±0.02 | 0.20 ±0.06 | 0.041 ±0.03 | 0.20 ±0.06 | 0.069 ±0.02 | 0.20 ±0.06 |

## B  Dataset Statistics

Table 3: Dataset Statistics.

|  | Number of Train users | Validation Users | Test users | Number of Items | Average rating | Sparsity | Number of Genres |
|---|---|---|---|---|---|---|---|
| ML-1M | 5,537 | 250 | 250 | 2,745 | 3.63 | 0.943 | 11 |
| Netflix | 7,978 | 1,000 | 1,000 | 3,081 | 3.60 | 0.904 | 15 |
| Goodbooks | 7,980 | 1,000 | 1,000 | 8,093 | 3.97 | 0.988 | 35 |

## C  Training Details

For our proposed models, we use the ADAMW optimizer while for AE models we use ADAM. For simplicity, we only refer to TEARS configurations, but note all configurations are the exact same for GERS. For TEARS models we train for 200 epochs using a batch size of 32, we do not use early stopping, but choose the best checkpoint across the 100 epochs. For TEARS Base the best checkpoint is chosen on NDCG@50 while for TEARS-VAEs we use the average NDCG@50 evaluated at $\alpha = \{0, 0.5, 1\}$. For AE models we train for 200 epochs with a batch size of 500. We choose the best checkpoint based on NDCG@50. TEARS models were trained using two Nvidia RTX-8000 GPU, with an average runtime of about 5 hours to complete, although we observe TEARS converges with much less than 200 epochs depending on the learning rate. AE models are trained using a single GPU and took on average 10-20 minutes (depending on the model) to complete.

- *TEARS*: For TEARS-VAEs and TEARS-Base, we tune dropout $\in \{0.1, 0.2, 0.4\}$ the learning rate (LR) $\in \{0.001, 0.0001\}$. Aditionally, For TEARS-VAEs we tune $\lambda_1 \in \{0.1, 0.5, 1\}$. We choose to not tune $\lambda_2$ and use an annealing schedule up to $\lambda_2 = 0.5$
- *Multi-VAE* : We tune dropout $\in \{0.1, 0.2, 0.4\}$, the learning rate $\in \{0.001, 0.0001, 0.00001\}$, $\beta \in \{0.1, 0.3, 0.5\}$ with a standard annealing schedule found in [30], and weight decay $\in \{0, 0.00001\}$.
- *Multi-DAE* We tune dropout $\in \{0.1, 0.2, 0.4\}$ and the learning rate $\in \{0.001, 0.0001, 0.00001\}$.
- *RecVAE* We tune dropout $\in \{0.1, 0.2, 0.4\}$, LR $\in \{0.001, 0.0001, 0.00001\}$, $\gamma \in \{.0035, .004, .005\}$ and a weight decay $\in \{0, 0.00001\}$. We additionally use the loss function provided by the authors [45], only for this model specifically .
- *MacridVAE* We tune dropout $\in \{0.1, 0.2, 0.4\}$, LR $\in \{0.001, 0.0001, 0.00001\}$, the number of concepts $k \in \{2, 4, 8, 16\}$ and a weight decay $\in \{0, 0.00001\}$.
- *EASE* We tune $\lambda$ over 50 values ranging between $[1, 10, 000]$ spread evenly. An additional detail is that only for this model, normalize ratings $r \in [0, 1]$, which yielded better results.

### C.1  TEARS-MacridVAE

MacridVAE decomposes the user representation into multiple disentangled concept representations which are normalized across the concept dimensions, thus, to be able to properly interpolate between the black-box and summary embeddings we do the same procedure for

TEARS-MacridVAE such that:

$$z_{u,r} = \left[ z_{u,r}^{(1)}, z_{u,r}^{(2)}, ..., z_{u,r}^{(K)} \right] \tag{10}$$

$$\mu_{\text{normalized},u,r}^{(k)} = \frac{\mu_{u,r}^{(k)}}{||\mu_{u,r}^{(k)}||_2} \tag{11}$$

$$z_{u,s} = \left[ z_{u,s}^{(1)}, z_{u,s}^{(2)}, ..., z_{u,s}^{(K)} \right] \tag{12}$$

$$\mu_{\text{normalized},u,s}^{(k)} = \frac{\mu_{u,s}^{(k)}}{||\mu_{u,s}^{(k)}||_2} \tag{13}$$

$$\tag{14}$$

Additionally, MacridVAE's first layer representations, often thought of as analogous to the item representations in AE recommender models, are shared with the last layer's representations. Since we freeze the encoder model at the beginning of training, we found that making a copy of MacridVAE's input representations, freezing them, and then allowing the final layers representations to be trained led to the best results and highest consistency in logic with other models.

# D    Few-Shot Prompting GPT for Recommendations

Table 5 displays the prompting strategy used to obtain GPT-4Turbo recommendation metrics in Table 1. We post-process the GPT output it is successfully in the requested format. If there is a failure we file the request again, for up to 10 times, after which we declare it as a failure and record the respective metrics as 0. We overall have 78 failures for the Netflix dataset and 24 failures for ML-1M. We do not include failures when calculating the metrics in 5, which is favorable for GPT.

**Table 4: Prompting strategy for few-shot recommendations.**

| Aspect | Value |
|---|---|
| Input Prompt | User summary: |
| | {user summary} |
| | Here are the available {item_type}: |
| | {All {item_type} in the catalog in ID: Title format} |
| | Important, do not recommend the following movies as they have already been seen by the user: |
| | {Seen {item_type} by user} |
| | Please only output the top 100 {item_type}. Simply print their id do not use the title output the movies in the format: id1, id2, ... idn} |

# E    GPT Netflix Results

We visualize the large-scope changes for the Netflix dataset in this appendix. Figure 12 shows the details. Our findings are consistent with those of §6.1 where we find TEARS-MacridVAE can consistently outperform TEARS-Base and TEARS-Multi-Vae performs poorest on controllability tasks.

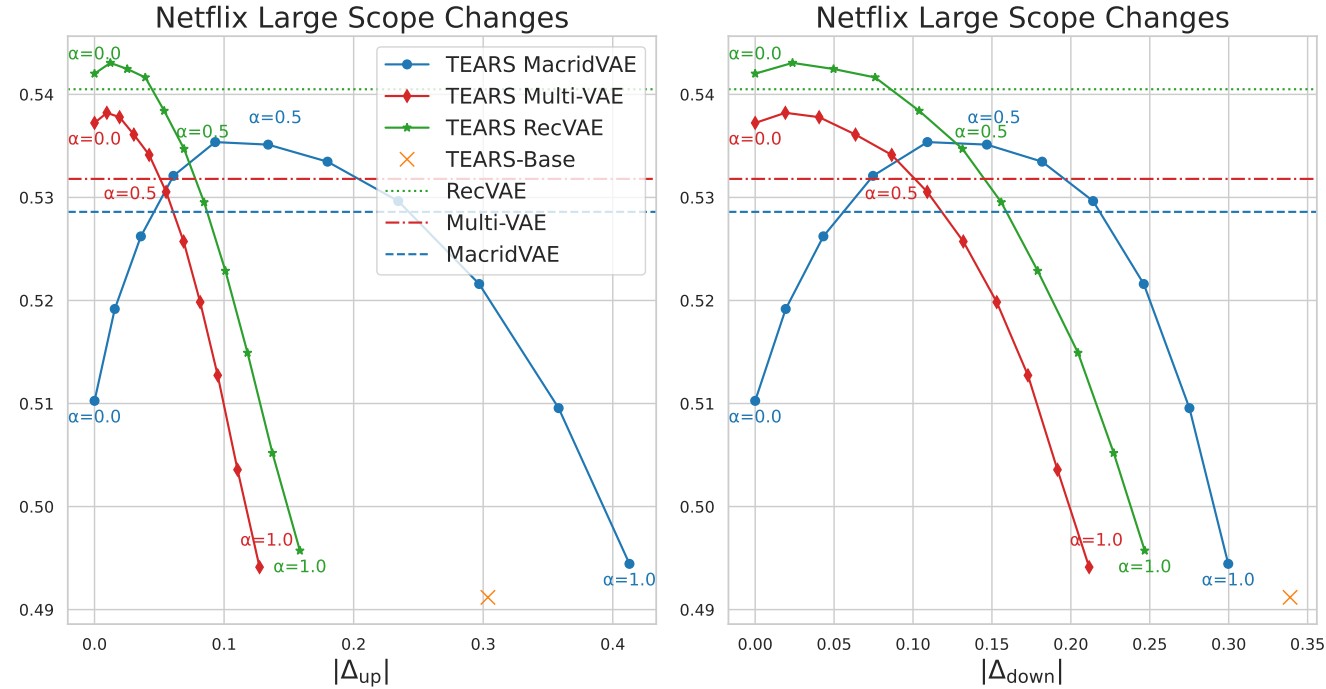

**Figure 6: Tradeoff between recommendation and controllability for the Netflix dataset. The x-axis represents $|\Delta_{\mathbf{up/down}}|$ as $\alpha$ decreases. We see consistent results for both LLMs with those observed in ML-1M and Goodbooks.**

## F    LLaMA 3.1 results

We report the controllability results for models trained using LLaMA-3.1 405b generated summaries. Importantly all textual edits for controllability are made by LLaMA-3.1 405b, we specifically set the seed and set the temperature to 0, this allows our results to be fully reproducible with an open weights model. As the edits are all made using LLaMA, we do not compare controllability results with GPT-based models, since results may be impacted by how good the editing LLM is at making edits, thus we cannot identify which model is better.

### F.1    Large Scope Changes & Guided Recommenations

Figure 7 illustrates the tradeoff between recommendation performance and controllability as $\alpha$ increases for ML-1M and Goodbooks datasets, with Figure E showing similar results for Netflix. These findings are consistent with those observed in GPT models, demonstrating that all models exhibit controllability. Notably, the TEARS VAE models consistently outperform TEARS Base at $\alpha = 1$ while maintaining comparable controllability, underscoring the advantages of our hybrid approach combined with the OT procedure.

In the guided recommendations experiment, we observe performance similar to that of GPT-based models, where short guiding phrases effectively steer user black-box embeddings, enabling users to contextualize their experience. This further validates the effectiveness of our approach across different model architectures and datasets.

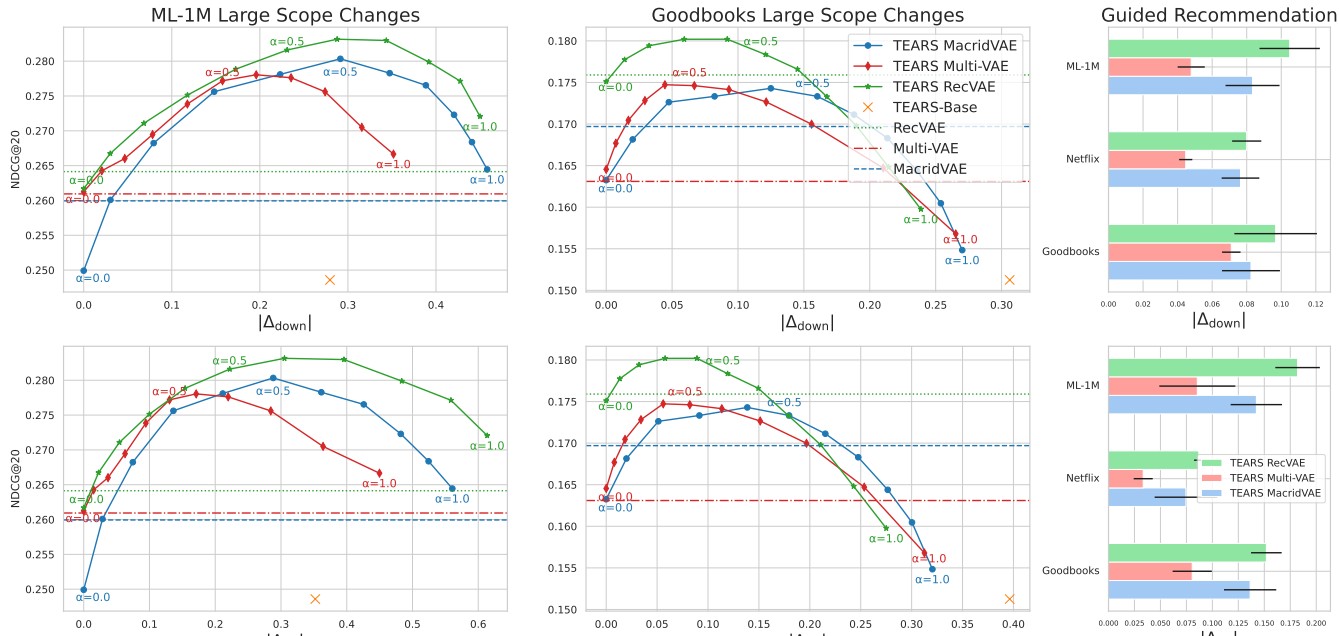

Figure 7: Tradeoff between recommendation performance and large scope controllability for ML-1M and Goodbooks.

## F.2 Netflix Results

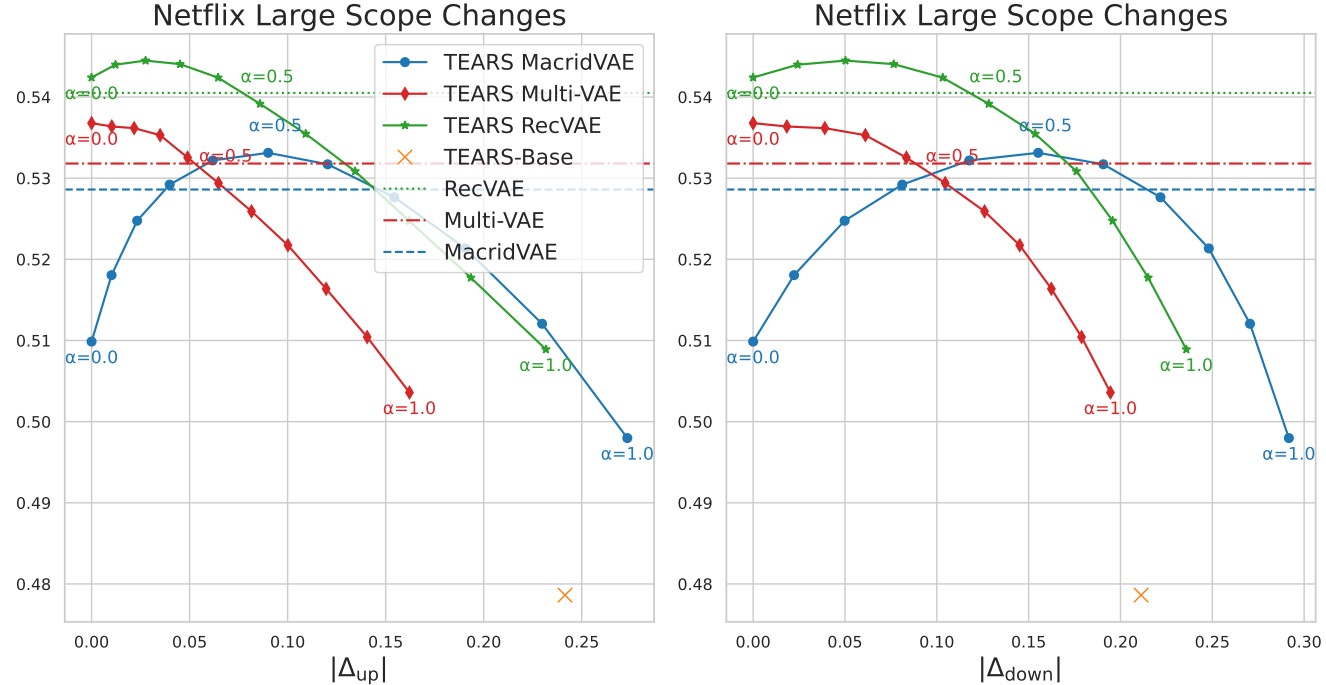

Figure 8: Tradeoff between recommendation and controllability for the Netflix dataset. The x-axis represents $|\Delta_{\text{up/down}}|$ as $\alpha$ decreases. We see consistent results for both LLMs with those observed in ML-1M and Goodbooks. We specifically observe how LLaMA models achieve better recommendation performance at higher values of $\alpha$.

## F.3 Small Scale Experiments

In this section, we replicate the fine-grained experiments using LLaMA-3.1 405b. To ensure full reproducibility, we set a specific random seed and adjust the model's temperature to 0. It's worth noting that we observed setting the temperature generally diminished the model's performance on the editing task, this is due to the model often returning the same response over multiple repetitions hindering the diversity of edits. Although we empirically found better results without temperature adjustment, we present here the most reproducible results. We found that TEARS MacridVAE in the Goodbooks dataset, performed significantly poorly with an average of $\delta_{\text{rank}} = -33.22$. We observed this was due to a low percentage of outliers that significantly lowered the mean to negative proportions. This is because our target items lie in the range of 100-500, making it so that items can go down a lot more than they can go up. Thus, only for this model in this one dataset, we report values that exclude the 0.075th smallest quantile which gives a less skewed representation of results. Overall, we observe that results are mostly consistent with those observed with GPT, being able to consistently increase the rank for all models (after adjusting for outliers) in all datasets.

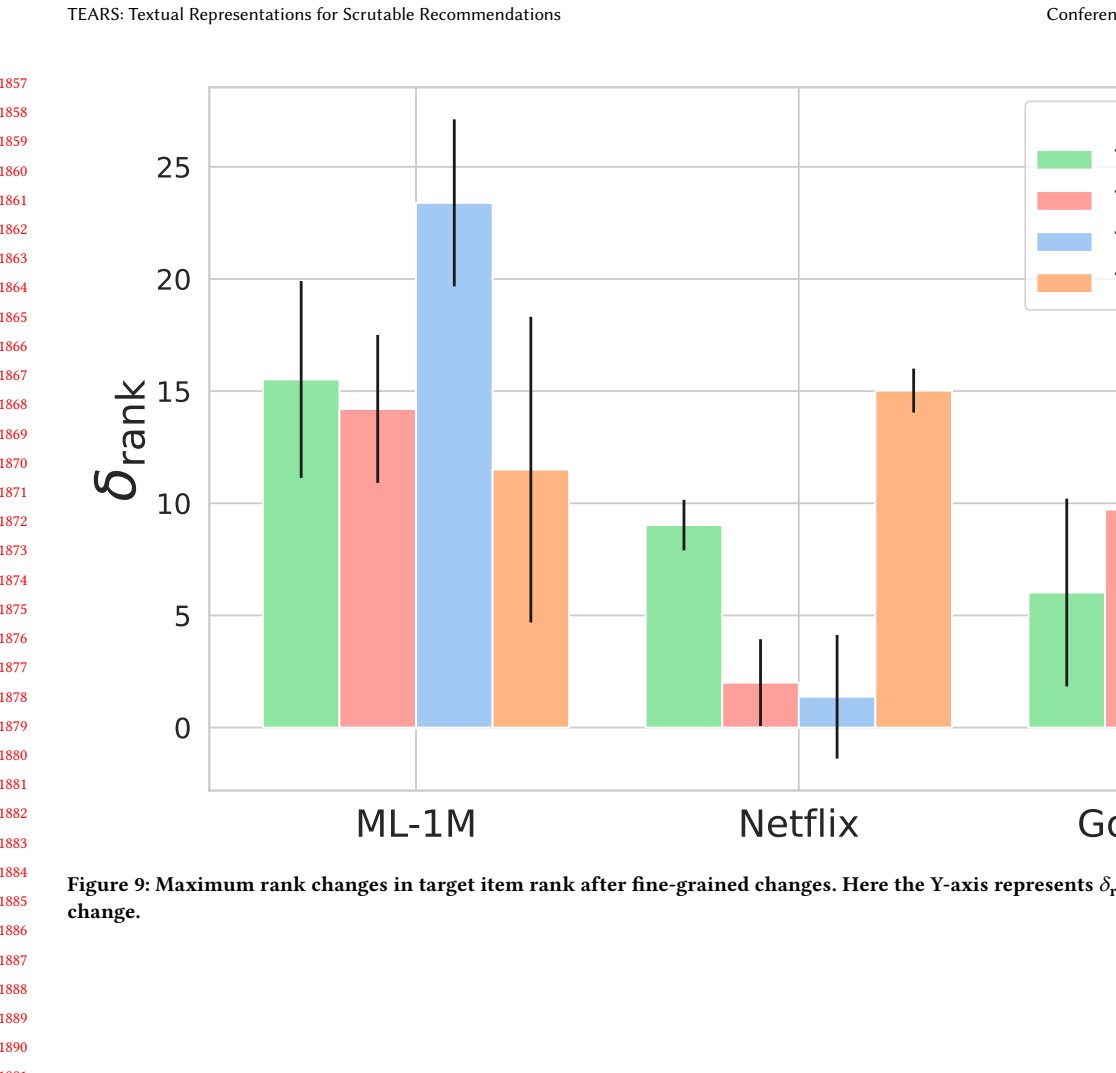

**Figure 9: Maximum rank changes in target item rank after fine-grained changes. Here the Y-axis represents $\delta_{\text{rank}}$ = original rank - rank after change.**

# G  Large-Scope Changes Example

| Aspect | Value |
|---|---|
| MovieLens ID | 929 |
| Input prompt | Summary: The user enjoys a blend of drama, romance, and inspirational true stories, often favoring narrative depth and emotional arcs that showcase personal growth, complex relationships, or historical significance. Plots that revolve around overcoming adversity, exploring human connections, or facing moral dilemmas tend to resonate well. The user also has a strong appreciation for action and war dramas where the stakes are high, and characters must showcase heroism or make profound sacrifices. However, the user does not enjoy certain horror and animated films, particularly those with a darker, more disturbing premise or themes that might be deemed as too intense or mature, and animated features that perhaps skew towards a younger audience or do not present as complex or emotionally engaging stories. Plots that incorporate extreme violence or psychological terror, as well as those that lack the depth of character development found in dramas, are generally less appreciated, despite their popularity among other audiences who may enjoy the thrill and suspense they provide. |
| Genre Identification Prompt | You are a professional editor. Please identify the user-preferred genres from the following: {genre_set} |
| Favourite Genre | Drama |
| Least Favourite Genre | Horror |
| User Prompt

Least Favorite: [genre] {user_summary}. | Please identify the user's most favorite genre from the following summary and the least favorite genre in the format Favorite: [genre] |
| Editing Prompt | Now using this setup, write a new summary in the same style that reflects that {favorite_genre} is your least favorite and {least_favorite_genre} is your favorite. Only output the full summary, keep the format and length the same. |
| Output | The user has a profound affinity for the suspenseful and chilling narratives that horror films provide, often seeking out stories that delve into the supernatural, psychological terror, or the macabre. They appreciate the adrenaline rush and the exploration of darker themes that challenge their fears and push the boundaries of comfort. The user is particularly drawn to horror films that cleverly twist reality and offer innovative scares or thought-provoking premises. They also enjoy the high stakes and intense scenarios found in action-packed horror where survival is a key element. Conversely, the user finds little interest in drama, as they tend to avoid films that focus heavily on emotional arcs, personal growth, and the intricacies of human relationships. They perceive these narratives as lacking the excitement and visceral engagement they crave. Dramas that emphasize historical significance or moral dilemmas do not captivate their attention in the same way that a thrilling, heart-pounding horror story does. The user is less moved by stories of overcoming adversity or exploring complex relationships unless they are framed within the context of a horror setting. |
| $\Delta_{\text{Down}}@50$ | 0.461 |
| $\Delta_{\text{Up}}@50$ | -0.730 |

## H  Fine Grained Experiments Supplemental

### H.1  Prompting Procedure for Fine-Grained Controllability

Table 5: Prompting Scheme for fine-grained experiment. We note we specifically ask for the identified words to be inserted into the summary together, but explicitly ask to replace a sentence in the summary.

| First Prompt | You are a professional editor. Please summarize the {item_type} into 5 words only, referring to plot points/themes: {item}. |
|---|---|
| Second Prompt | Now, using this, edit those 5 words into this summary. Replace a sentence where it makes sense. Only output the summary. {summary} only output the new summary, making sure the 5 new comma-separated words are in a new sentence, replacing an old one somewhere together in the new summary. |

### H.2  Breakdown by value of $\alpha$

Figure 10 visualizes the relationship between the fine grained changes and the value of $\alpha$. We generally observe that higher levels of alpha lead to higher levels of controllability, with some exceptions. Importantly we see that for all models in all datasets at all $\alpha$s we observe a positive $\delta_{\text{rank}}$. The procedure for producing these for LLaMA is discussed in App. 10b in detail.

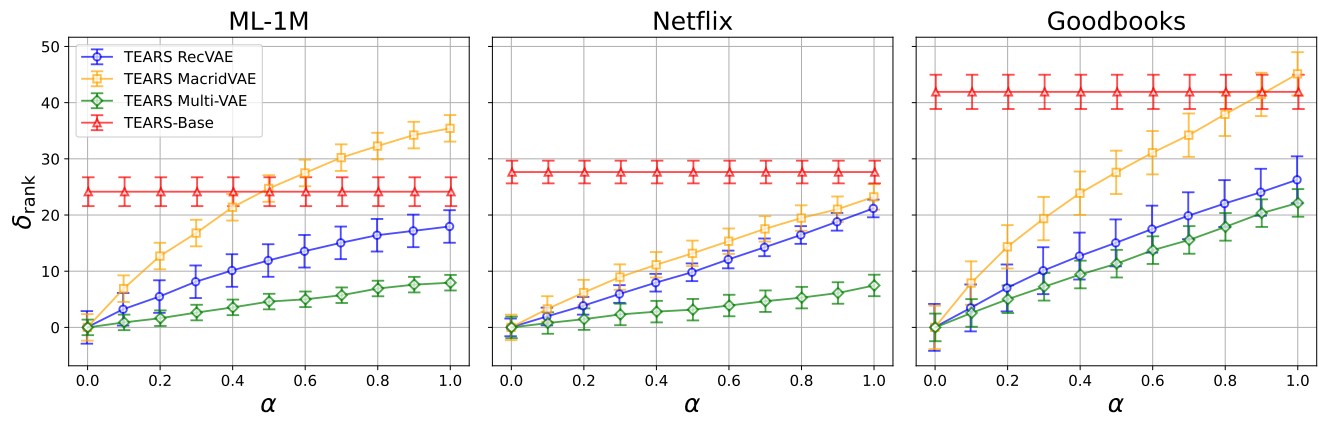

(a) GPT

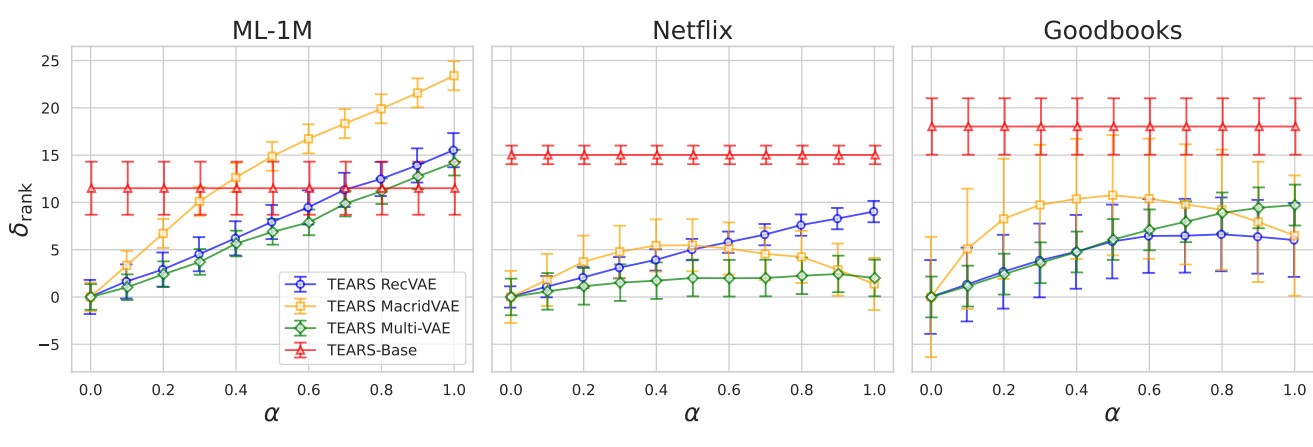

(b) LLaMA

Figure 10: $\delta_{\text{rank}}$ broken down by $\alpha$ for each dataset with error bars representing the standard error. We observe for all models there is a value of $\alpha$ for which we are able to increase the rank of the target item.

## H.3 ML-1M Examples

| Dataset | ML-1M |
|---|---|
| Original Summary | Summary: The user enjoys a variety of genres with a strong preference for comedy, often blended with elements of romance, drama, and action. Bent towards comedies that deliver a mix of witty dialogue, quirky characters, and situations that lead to both heartwarming and humorous outcomes is evident. The user appreciates horror when it is juxtaposed with humor, and thrillers that contain supernatural or fantastical elements are particularly enjoyable. **Storylines involving personal growth, unconventional relationships, and comedic misadventures also resonate well.** 

 Conversely, the user does not enjoy certain actions and science fiction films as much, especially if they lack a comedic element or deeper narrative. Plots that focus heavily on conventional action sequences, with less emphasis on character development or innovative storytelling, are less favorable. The user might be less interested in sci-fi adventures that are more serious and lacking the playful or satirical tone found in more favored titles. While tension and suspense are appreciated in certain contexts, straightforward action-driven thrillers without substantial plot twists or character complexity may not capture the user's interest. |
| Augmented Summary | The user enjoys a variety of genres with a strong preference for comedy, often blended with elements of romance, drama, and action. Bent towards comedies that deliver a mix of witty dialogue, quirky characters, and situations that lead to both heartwarming and humorous outcomes is evident. The user appreciates horror when it is juxtaposed with humor, and thrillers that contain supernatural or fantastical elements are particularly enjoyable. **Self-worth, community, sacrifice, redemption, family resonate well.** 

 Conversely, the user does not enjoy certain actions and science fiction films as much, especially if they lack a comedic element or deeper narrative. Plots that focus heavily on conventional action sequences, with less emphasis on character development or innovative storytelling, are less favorable. The user might be less interested in sci-fi adventures that are more serious and lacking the playful or satirical tone found in more favored titles. While tension and suspense are appreciated in certain contexts, straightforward action-driven thrillers without substantial plot twists or character complexity may not capture the user's interest. |
| Target Item | *It's a Wonderful Life* (1946) |
| Original Rank | 259 |
| New Rank | 235 |
| $\delta_{\text{rank}}$ | 24 |
| Dataset | ML-1M |
| Original Summary | Summary: The user has a clear preference for genres that blend comedy with other elements, such as sci-fi, horror, and action. They particularly enjoy comedic films that explore the dynamic interplay between humor and speculative fiction, most likely appreciating how these genres can satirize or comment on society and our relationship with technology. **The user also gravitates towards dramas that are infused with sci-fi and adventure, often valuing intricate plots that weave in elements of thrill and suspense, and possibly favoring storylines that involve exploration, the supernatural, and high stakes situations.** 

 Conversely, the user does not enjoy animations as much, especially those targeted primarily at children. This suggests a lesser interest in stories that are perceived as being too simplistic or juvenile. Furthermore, the user seems disinterested in musicals and fantastical adventures that prioritize whimsy over mature humor or complex storytelling. While other users might appreciate the innocence and escapism offered by these genres, this user shows a predilection for more sophisticated narratives found in films that provide a mix of laughter with thought-provoking or action-driven content. |
| Augmented Summary | Summary: The user has a clear preference for genres that blend comedy with other elements, such as sci-fi, horror, and action. They particularly enjoy comedic films that explore the dynamic interplay between humor and speculative fiction, most likely appreciating how these genres can satirize or comment on society and our relationship with technology. **Sports agent's redemption through love.** 

 Conversely, the user does not enjoy animations as much, especially those targeted primarily at children. This suggests a lesser interest in stories that are perceived as being too simplistic or juvenile. Furthermore, the user seems disinterested in musicals and fantastical adventures that prioritize whimsy over mature humor or complex storytelling. While other users might appreciate the innocence and escapism offered by these genres, this user shows a predilection for more sophisticated narratives found in films that provide a mix of laughter with thought-provoking or action-driven content. |
| Target Item | *Jerry Maguire* (1996) |
| Original Rank | 136 |
| New Rank | 108 |
| $\delta_{\text{rank}}$ | 28 |

## I    Guided Recommendation

We further visualize the process of generating guided recommendations for three different genres in the ML-1M dataset using TEARS RecVAE. To accomplish this, we employ t-SNE [51] to visualize two types of embeddings: the mean latent of the black-box embeddings (displayed in red) and the mean latent for the text embeddings (displayed with a color gradient). Our observations reveal that guiding the recommendations has a personalized effect for each user. Individual user representations move towards the genre representation in unique ways. This personalization can be attributed to changes in recommendations that suggest items belonging to the target genre while still aligning with the individual user's preferences.

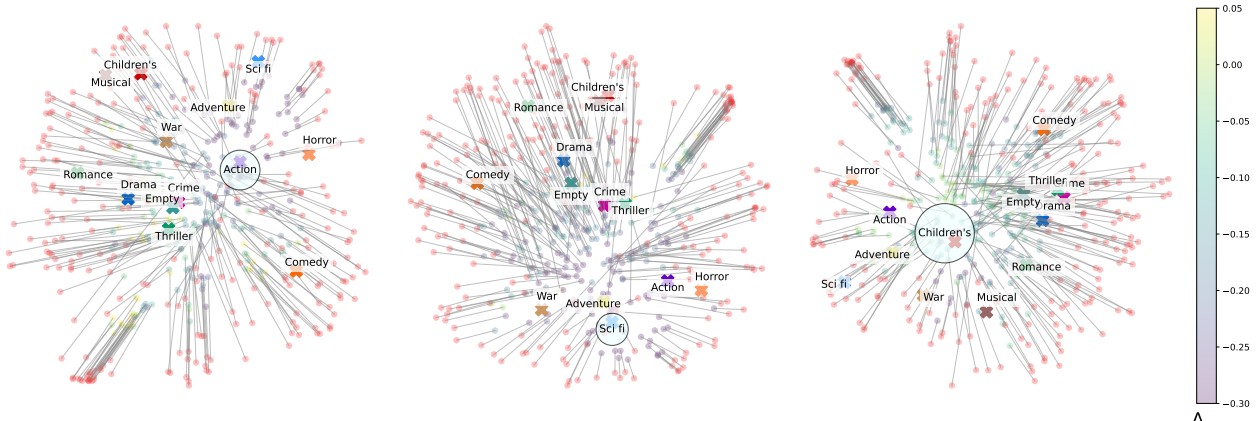

Figure 11: Rank changes in target item rank after fine grained changes. Y-axis represents $\delta_{\text{rank}}$ = New rank - Old Rank.

## J    Genre Based Controllability

Since GERS represents users through a combination of black-box embeddings and a genre-based vector representation, a direct comparison with TEARS becomes challenging. The controllability experiments we propose focus on text-based modifications, not genre vectors as used in GERS. However, we outline an approach to analyze GERS's controllability and compare it with TEARS.

We aim to replicate the large-scale experiments described in §6.1, where user interests are drastically shifted. To do this with GERS, we simulate a similar scenario by assigning the proportion of the user's least favorite genre to that of their original favorite genre, and vice-versa. This adjustment mimics the interest flip applied in TEARS. However, unlike TEARS, this approach is less coarse-grained, as the summaries may reflect varying levels of interest after the change. Since the shift is not perfectly mirrored, we expect GERS to be at a disadvantage compared to TEARS, as the summaries in GERS are not tied to a single metric but can express a broad spectrum of genre preferences. Despite this limitation, we compare this setup to the original TEARS results, measuring performance using $\Delta@k$.

While the described setting may disadvantage GERS, we can also construct a scenario where it acts as an upper bound for controllability performance. For example, when flipping user interests, we could zero out all genres except for the least favorite, assigning it full relevance in the user profile. This simulation acts like a genre-specific filter while still accounting for popularity biases in the dataset. In contrast, we do not expect TEARS to achieve this level of control, as its summaries express a mix of interests, even after edits. We also note one would not want TEARS to reach this level of controllability, as the purpose of the summaries is to pose an accurate textual representation of the user's interests not as a filtering mechanism simple heuristics may be more suitable for. We consider this scenario an upper-bound for controllability, and refer to this measure as $\Delta@k_{\text{upper}}$.

In Table 6, we present the values of $\Delta@k$ and $\Delta@k_{\text{upper}}$ averaged over five seeds for all datasets. We observe that TEARS models generally perform better on $\Delta@k$ because they can more accurately describe shifts in user interests across different genres, while GERS models are more constrained in this specific setting. Additionally, we find that $\Delta@k_{\uparrow}$ is often two to three times higher than $\Delta@k$ for TEARS models, indicating that TEARS does not simply flip the user's interest toward one genre, but also accounts for related genres, themes, and plot points. This suggests that TEARS can properly adjust user preferences and influence the ranking outcomes, while still maintaining a balance by considering other relevant interests. In contrast, GERS, with its genre-specific focus, may lack the flexibility TEARS offers in representing the complexity of user preferences, especially when the user has to adjust a large number of genres to properly portray their preferences (i.e Goodbooks with 39 genres).

**Table 6: Controllability Analysis of TEARS and GERS models on ML-1M, Netflix, and Goodbooks datasets. For TEARS, we report $\Delta@k$, as described in §6.1, with some adjustments made to accommodate GERS. We also report $\Delta@k_\uparrow$, representing the case of flipping a user's interest and assigning full weight to a single genre in GERS. TEARS strikes a balance between the two approaches, indicating its ability to adjust user preferences without overly relying on a single genre.**

| Dataset | Model | $\Delta@20_{up,\alpha=1}$ | $\Delta@20_{down,\alpha=1}$ | $\Delta@20_{up,\alpha=0.5}$ | $\Delta@20_{down,\alpha=0.5}$ | $\Delta@20_{up,\uparrow,\alpha=1}$ | $\Delta@20_{down,\uparrow,\alpha=1}$ | $\Delta@20_{up,\uparrow,\alpha=0.5}$ | $\Delta@20_{down,\uparrow,\alpha=0.5}$ |
|---|---|---|---|---|---|---|---|---|---|
| ML-1M | TEARS Base | $0.564 \pm 0.088$ | $0.344 \pm 0.049$ | N/A | N/A | N/A | N/A | N/A | N/A |
| | GERS Base | $0.220 \pm 0.007$ | $0.168 \pm 0.005$ | N/A | N/A | $0.696 \pm 0.020$ | $0.319 \pm 0.012$ | $0.696 \pm 0.020$ | $0.319 \pm 0.012$ |
| | TEARS RecVAE | $0.417 \pm 0.043$ | $0.243 \pm 0.026$ | $0.175 \pm 0.011$ | $0.115 \pm 0.010$ | N/A | N/A | N/A | N/A |
| | GERS RecVAE | $0.271 \pm 0.017$ | $0.261 \pm 0.018$ | $0.113 \pm 0.010$ | $0.112 \pm 0.010$ | $0.770 \pm 0.004$ | $0.399 \pm 0.004$ | $0.747 \pm 0.005$ | $0.340 \pm 0.003$ |
| Netflix | TEARS Base | $0.564 \pm 0.088$ | $0.344 \pm 0.049$ | N/A | N/A | N/A | N/A | N/A | N/A |
| | GERS Base | $0.220 \pm 0.007$ | $0.168 \pm 0.005$ | N/A | N/A | $0.696 \pm 0.020$ | $0.319 \pm 0.012$ | $0.696 \pm 0.020$ | $0.319 \pm 0.012$ |
| | TEARS RecVAE | $0.417 \pm 0.043$ | $0.243 \pm 0.026$ | $0.175 \pm 0.011$ | $0.115 \pm 0.010$ | N/A | N/A | N/A | N/A |
| | GERS RecVAE | $0.271 \pm 0.017$ | $0.261 \pm 0.018$ | $0.113 \pm 0.010$ | $0.112 \pm 0.010$ | $0.770 \pm 0.004$ | $0.399 \pm 0.004$ | $0.747 \pm 0.005$ | $0.340 \pm 0.003$ |
| Goodbooks | TEARS Base | $0.564 \pm 0.088$ | $0.344 \pm 0.049$ | N/A | N/A | N/A | N/A | N/A | N/A |
| | GERS Base | $0.220 \pm 0.007$ | $0.168 \pm 0.005$ | N/A | N/A | $0.696 \pm 0.020$ | $0.319 \pm 0.012$ | $0.696 \pm 0.020$ | $0.319 \pm 0.012$ |
| | TEARS RecVAE | $0.417 \pm 0.043$ | $0.243 \pm 0.026$ | $0.175 \pm 0.011$ | $0.115 \pm 0.010$ | N/A | N/A | N/A | N/A |
| | GERS RecVAE | $0.271 \pm 0.017$ | $0.261 \pm 0.018$ | $0.113 \pm 0.010$ | $0.112 \pm 0.010$ | $0.770 \pm 0.004$ | $0.399 \pm 0.004$ | $0.747 \pm 0.005$ | $0.340 \pm 0.003$ |

## K  Controllability Breakdown

Table 7 shows the controllability results for the large-scope and guided recommendation experiments averaged over five different seeds. Overall, we observe TEARS MacridVAE consistently outperforms other models and even TEARS BASE when it comes to controllability at an $\alpha = 1$. Overall, we find TEARS MacridVAE to be the best-performing model, having better recommendation performance than baselines for some value of $\alpha$ in all datasets while also excelling in the controllability tasks.

**Table 7: Comparison of controllability performance across different datasets and models. Each model is evaluated using five different seeds.**

| Dataset | Model Type | Model | Best $\alpha$ | Large Scope $|\Delta_{up,\alpha=1}|$ | Large Scope $|\Delta_{up,\alpha=0.5}|$ | Large Scope $|\Delta_{down,\alpha=1}|$ | Large Scope $|\Delta_{down,\alpha=0.5}|$ | Genre $|\Delta_{up,\alpha=0.5}|$ | Genre $|\Delta_{down,\alpha=0.5}|$ |
|---|---|---|---|---|---|---|---|---|---|
| ML-1M | GPT | TEARS Multi-VAE | 0.425 | $0.201 \pm 0.008$ | $0.072 \pm 0.015$ | $0.193 \pm 0.005$ | $0.085 \pm 0.015$ | $-0.068 \pm 0.004$ | $0.051 \pm 0.008$ |
| | | TEARS MacridVAE | 0.500 | $0.690 \pm 0.052$ | $0.357 \pm 0.036$ | $0.453 \pm 0.029$ | $0.250 \pm 0.021$ | $-0.272 \pm 0.019$ | $0.074 \pm 0.023$ |
| | | TEARS RecVAE | 0.425 | $0.391 \pm 0.013$ | $0.150 \pm 0.021$ | $0.346 \pm 0.012$ | $0.165 \pm 0.003$ | $-0.213 \pm 0.013$ | $0.101 \pm 0.013$ |
| | | TEARS Base | N/A | $0.387 \pm 0.104$ | $0.387 \pm 0.104$ | $0.294 \pm 0.044$ | $0.294 \pm 0.044$ | N/A | N/A |
| | LLaMA | TEARS MacridVAE | 0.50 | $0.561 \pm 0.058$ | $0.288 \pm 0.032$ | $0.458 \pm 0.038$ | $0.291 \pm 0.014$ | $-0.142 \pm 0.016$ | $0.083 \pm 0.025$ |
| | | TEARS Multi-VAE | 0.48 | $0.450 \pm 0.097$ | $0.130 \pm 0.025$ | $0.352 \pm 0.068$ | $0.158 \pm 0.028$ | $-0.086 \pm 0.008$ | $0.048 \pm 0.037$ |
| | | TEARS RecVAE | 0.54 | $0.614 \pm 0.029$ | $0.222 \pm 0.011$ | $0.450 \pm 0.018$ | $0.231 \pm 0.009$ | $-0.182 \pm 0.017$ | $0.105 \pm 0.021$ |
| | | TEARS Base | N/A | $0.352 \pm 0.155$ | $0.352 \pm 0.155$ | $0.280 \pm 0.074$ | $0.280 \pm 0.074$ | N/A | N/A |
| Netflix | GPT | TEARS Multi-VAE | 0.140 | $0.064 \pm 0.013$ | $0.026 \pm 0.005$ | $0.045 \pm 0.013$ | $0.026 \pm 0.006$ | $-0.039 \pm 0.004$ | $0.024 \pm 0.007$ |
| | | TEARS MacridVAE | 0.520 | $0.329 \pm 0.060$ | $0.093 \pm 0.009$ | $0.071 \pm 0.026$ | $0.032 \pm 0.018$ | $-0.073 \pm 0.028$ | $0.085 \pm 0.062$ |
| | | TEARS RecVAE | 0.160 | $0.094 \pm 0.023$ | $0.034 \pm 0.009$ | $0.070 \pm 0.011$ | $0.038 \pm 0.006$ | $-0.044 \pm 0.010$ | $0.064 \pm 0.031$ |
| | | TEARS Base | N/A | $0.219 \pm 0.031$ | $0.219 \pm 0.031$ | $0.119 \pm 0.015$ | $0.119 \pm 0.015$ | N/A | N/A |
| | LLaMA | TEARS MacridVAE | 0.54 | $0.273 \pm 0.064$ | $0.090 \pm 0.011$ | $0.292 \pm 0.035$ | $0.155 \pm 0.010$ | $-0.075 \pm 0.011$ | $0.076 \pm 0.030$ |
| | | TEARS Multi-VAE | 0.14 | $0.163 \pm 0.005$ | $0.065 \pm 0.001$ | $0.195 \pm 0.008$ | $0.105 \pm 0.005$ | $-0.033 \pm 0.004$ | $0.045 \pm 0.009$ |
| | | TEARS RecVAE | 0.32 | $0.232 \pm 0.022$ | $0.086 \pm 0.002$ | $0.236 \pm 0.014$ | $0.128 \pm 0.009$ | $-0.087 \pm 0.008$ | $0.080 \pm 0.005$ |
| | | TEARS Base | N/A | $0.241 \pm 0.022$ | $0.241 \pm 0.022$ | $0.211 \pm 0.010$ | $0.211 \pm 0.010$ | N/A | N/A |
| Goodbooks | GPT | TEARS Multi-VAE | 0.380 | $0.394 \pm 0.054$ | $0.122 \pm 0.015$ | $0.237 \pm 0.024$ | $0.074 \pm 0.006$ | $-0.070 \pm 0.006$ | $0.050 \pm 0.016$ |
| | | TEARS MacridVAE | 0.380 | $0.597 \pm 0.024$ | $0.328 \pm 0.014$ | $0.318 \pm 0.023$ | $0.156 \pm 0.015$ | $-0.166 \pm 0.015$ | $0.122 \pm 0.005$ |
| | | TEARS RecVAE | 0.160 | $0.417 \pm 0.043$ | $0.175 \pm 0.011$ | $0.243 \pm 0.026$ | $0.115 \pm 0.010$ | $-0.211 \pm 0.009$ | $0.137 \pm 0.015$ |
| | | TEARS Base | N/A | $0.564 \pm 0.088$ | $0.564 \pm 0.088$ | $0.344 \pm 0.049$ | $0.344 \pm 0.049$ | N/A | N/A |
| | LLaMA | TEARS MacridVAE | 0.32 | $0.320 \pm 0.046$ | $0.180 \pm 0.019$ | $0.270 \pm 0.032$ | $0.160 \pm 0.016$ | $-0.136 \pm 0.017$ | $0.082 \pm 0.025$ |
| | | TEARS Multi-VAE | 0.50 | $0.313 \pm 0.033$ | $0.082 \pm 0.003$ | $0.265 \pm 0.020$ | $0.067 \pm 0.003$ | $-0.081 \pm 0.005$ | $0.071 \pm 0.019$ |
| | | TEARS RecVAE | 0.22 | $0.275 \pm 0.047$ | $0.119 \pm 0.011$ | $0.239 \pm 0.023$ | $0.121 \pm 0.007$ | $-0.152 \pm 0.024$ | $0.097 \pm 0.015$ |
| | | TEARS Base | N/A | $0.396 \pm 0.027$ | $0.396 \pm 0.027$ | $0.306 \pm 0.025$ | $0.306 \pm 0.025$ | N/A | N/A |

## L  Analysis Using $\alpha = 1$

We analyze the performance of TEARS and GERS variants using $\alpha = 1$ across all assessed recommendation metrics. Table 8 shows the recommendation performance for all LLaMA- and GPT-based models. We find TEARS RecVAE to consistently be the best-performing TEARS variant, outperforming other TEARS models. Notably, the performance boost with $\alpha = 1$ is not exclusive to TEARS; GERS RecVAE also demonstrates improved performance compared to GERS Base. We observe that that wether TEARS RecVAE or GERS RecVAE performs better largely on the dataset. GERS RecVAE performs better on Goodbooks, likely due to the higher number of genres that allow for more fine-grained specifications, while TEARS RecVAE performs best on ML-1M, which has fewer genres where user summaries being able to give more coarse-grained descriptions of preferences is more effective. Both models perform similarly on Netflix, a result consistent with §5.2, likely because the summaries are predominantly genre-based.

Table 8: Performance comparison of different TEARS models across ML-1M, Netflix, and Goodbooks datasets, separated by LLM (GPT and LLaMA).

| Dataset | Model | Recall@20 | NDCG@20 | Recall@50 | NDCG@50 |
|---|---|---|---|---|---|
| ML-1M | 🟢 TEARS Base | $0.267 \pm 0.004$ | $0.253 \pm 0.002$ | $0.302 \pm 0.014$ | $0.250 \pm 0.005$ |
| | ∞ TEARS Base | $0.259 \pm 0.010$ | $0.249 \pm 0.010$ | $0.292 \pm 0.015$ | $0.245 \pm 0.010$ |
| | 🟢 TEARS Multi-VAE$_{\alpha=1}$ | $0.268 \pm 0.007$ | $0.253 \pm 0.006$ | $0.290 \pm 0.005$ | $0.247 \pm 0.004$ |
| | ∞ TEARS Multi-VAE$_{\alpha=1}$ | $0.285 \pm 0.006$ | $0.267 \pm 0.004$ | $0.317 \pm 0.004$ | $0.264 \pm 0.003$ |
| | 🟢 TEARS Macrid VAE$_{\alpha=1}$ | $0.296 \pm 0.004$ | $0.264 \pm 0.004$ | $0.343 \pm 0.003$ | $0.268 \pm 0.003$ |
| | ∞ TEARS Macrid VAE$_{\alpha=1}$ | $0.294 \pm 0.003$ | $0.264 \pm 0.003$ | $0.344 \pm 0.005$ | $0.269 \pm 0.004$ |
| | 🟢 TEARS RecVAE$_{\alpha=1}$ | $0.293 \pm 0.005$ | $0.262 \pm 0.002$ | $0.336 \pm 0.008$ | $0.266 \pm 0.003$ |
| | ∞ TEARS RecVAE$_{\alpha=1}$ | $\mathbf{0.307 \pm 0.006}$ | $\mathbf{0.272 \pm 0.005}$ | $\mathbf{0.351 \pm 0.007}$ | $\mathbf{0.276 \pm 0.005}$ |
| | GERS RecVAE$_{\alpha=1}$ | $0.282 \pm 0.004$ | $0.258 \pm 0.003$ | $0.336 \pm 0.006$ | $0.262 \pm 0.002$ |
| Netflix | 🟢 TEARS Base | $0.465 \pm 0.004$ | $0.491 \pm 0.004$ | $0.413 \pm 0.003$ | $0.439 \pm 0.003$ |
| | ∞ TEARS Base | $0.452 \pm 0.002$ | $0.479 \pm 0.002$ | $0.397 \pm 0.001$ | $0.424 \pm 0.001$ |
| | 🟢 TEARS Multi-VAE$_{\alpha=1}$ | $0.468 \pm 0.001$ | $0.494 \pm 0.001$ | $0.414 \pm 0.002$ | $0.441 \pm 0.001$ |
| | ∞ TEARS Multi-VAE$_{\alpha=1}$ | $0.477 \pm 0.002$ | $0.504 \pm 0.002$ | $0.422 \pm 0.001$ | $0.449 \pm 0.001$ |
| | 🟢 TEARS Macrid VAE$_{\alpha=1}$ | $0.470 \pm 0.002$ | $0.494 \pm 0.001$ | $0.415 \pm 0.002$ | $0.441 \pm 0.001$ |
| | ∞ TEARS Macrid VAE$_{\alpha=1}$ | $0.475 \pm 0.004$ | $0.498 \pm 0.004$ | $0.421 \pm 0.003$ | $0.446 \pm 0.003$ |
| | 🟢 TEARS RecVAE$_{\alpha=1}$ | $0.471 \pm 0.002$ | $0.496 \pm 0.002$ | $0.418 \pm 0.002$ | $0.444 \pm 0.002$ |
| | ∞ TEARS RecVAE$_{\alpha=1}$ | $0.483 \pm 0.002$ | $\mathbf{0.509 \pm 0.001}$ | $\mathbf{0.428 \pm 0.002}$ | $\mathbf{0.455 \pm 0.001}$ |
| | GERS RecVAE$_{\alpha=1}$ | $\mathbf{0.485 \pm 0.001}$ | $\mathbf{0.509 \pm 0.001}$ | $\mathbf{0.428 \pm 0.001}$ | $0.454 \pm 0.001$ |
| Goodbooks | 🟢 TEARS Base | $0.143 \pm 0.002$ | $0.151 \pm 0.003$ | $0.157 \pm 0.004$ | $0.151 \pm 0.004$ |
| | ∞ TEARS Base | $0.143 \pm 0.002$ | $0.151 \pm 0.003$ | $0.156 \pm 0.002$ | $0.151 \pm 0.002$ |
| | 🟢 TEARS Multi-VAE$_{\alpha=1}$ | $0.151 \pm 0.002$ | $0.160 \pm 0.001$ | $0.160 \pm 0.003$ | $0.157 \pm 0.002$ |
| | ∞ TEARS Multi-VAE$_{\alpha=1}$ | $0.147 \pm 0.002$ | $0.157 \pm 0.002$ | $0.158 \pm 0.003$ | $0.155 \pm 0.002$ |
| | 🟢 TEARS Macrid-VAE$_{\alpha=1}$ | $0.152 \pm 0.003$ | $0.159 \pm 0.002$ | $0.164 \pm 0.001$ | $0.158 \pm 0.001$ |
| | ∞ TEARS Macrid-VAE$_{\alpha=1}$ | $0.147 \pm 0.001$ | $0.155 \pm 0.001$ | $0.161 \pm 0.001$ | $0.155 \pm 0.001$ |
| | 🟢 TEARS RecVAE$_{\alpha=1}$ | $0.152 \pm 0.002$ | $0.161 \pm 0.002$ | $0.166 \pm 0.001$ | $0.161 \pm 0.001$ |
| | ∞ TEARS RecVAE$_{\alpha=1}$ | $0.150 \pm 0.002$ | $0.160 \pm 0.003$ | $0.163 \pm 0.001$ | $0.159 \pm 0.001$ |
| | GERS RecVAE$_{\alpha=1}$ | $\mathbf{0.156 \pm 0.002}$ | $\mathbf{0.165 \pm 0.001}$ | $\mathbf{0.169 \pm 0.001}$ | $\mathbf{0.163 \pm 0.001}$ |

## M  Ablations

We perform a variety of ablations on the ML-1M dataset to assess the efficacy of the proposed method. All methods are assessed using the same random seed and the hyperparameters from the best-performing TEARS MacridVAE model, using the GPT-4-turbo generated summaries.

### M.1  Pooling and Optimal Transport

We compare mean-pooling with concatenation, another popular pooling method [56]. We assess $\text{NDCG}_s$, based on text embeddings, and $\text{NDCG}_c$, where mean-pooling uses $\alpha = 0.5$, while concatenation applies an MLP to map embeddings to the correct dimensions for recommendations. For controllability, we measure $\Delta_{\text{up}}@20$ and $\Delta_{\text{down}}@20$ with recommendations generated purely from $Z_s$ (yielding the best controllability). Additionally, we evaluate whether the OT objective is beneficial. Table 9 shows mean pooling without OT underperforms and is less controllable. We also find that concatenation performs poorly overall, with OT harming its performance but improving text-based results. Furthermore, concatenation without OT significantly reduces controllability. These results highlight that combining black-box and text embeddings improves both controllability and performance, with OT being crucial for enhancing controllability.

**Table 9: Ablation on different pooling and optimization strategies. We find the mean w OT is the most optimal in both recommendation performance and controllability.**

|  | NDCG@50$_c$ | NDCG@50$_s$ | $|\Delta_{\text{down}}@20|$ | $|\Delta_{\text{up}}@20|$ |
|---|---|---|---|---|
| Mean w OT | **0.296** | **0.273** | **0.470** | **0.740** |
| Mean w.o OT | 0.291 | 0.251 | 0.261 | 0.410 |
| Concat w OT | 0.259 | 0.269 | 0.425 | 0.610 |
| Concat w.o OT | 0.278 | 0.230 | 0.185 | 0.063 |

## M.2 Loss Function Configurations

Table 10 presents various configurations for optimizing $\mathcal{L}_R$. In practice, we define $\mathcal{L}_R = \mathcal{L}_r + \mathcal{L}_c + \mathcal{L}_s$, optimizing for recommendations based on black-box representations, summary representations, and a combination of both. Our goal is to evaluate the impact of these components on performance and controllability.

We observe that $\mathcal{L}_s$ plays a crucial role in enhancing the system's controllability, while $\mathcal{L}_c$ emerges as the key factor for performance, with performance dropping significantly in its absence. Notably, the model trained without $\mathcal{L}_s$ achieves the best performance when $\alpha = 0$, although it shows the lowest controllability. Interestingly, all configurations maintain some level of controllability, but the combination of all three losses provides the best balance between recommendation performance and controllability.

**Table 10: Comparison of metrics across different configurations.**

| $\mathcal{L}_r$ | $\mathcal{L}_c$ | $\mathcal{L}_s$ | NDCG@50$_r$ | NDCG@50$_c$ | NDCG@50$_s$ | $|\Delta_{\text{up}}@20|$ |
|---|---|---|---|---|---|---|
| x | x | x | 0.266 | **0.296** | **0.273** | **0.740** |
| x | x |  | **0.270** | 0.294 | **0.273** | 0.523 |
|  | x | x | 0.265 | 0.292 | 0.267 | 0.739 |
| x |  | x | 0.266 | 0.288 | 0.269 | 0.609 |

## M.3 What Weights to Train

We run ablations on what weights one should and should not update when training TEARS. Table 11 showcases different combinations of training regimens. An x here indicates that the model encoder weights are trained, for all methods we train decoder weights. Interestingly we find that when training both models, instabilities seem to arise not allowing the model to converge properly and yielding both worse recommendations and no controllability. Furthermore we observe that keeping the text encoder frozen but training the AE yields improved recommendations and some controllability, we imagine in this case, the AE is learning to more closely align to the text-encoders representations. Finally, our proposed training regimen of only updating the text-encoder's weights outperforms the prior two methods.

**Table 11: Performance metrics based on training components.**

| Train AE-encoder | Train text-encoder | NDCG@50 | $|\Delta_{\text{up}}@20|$ |
|---|---|---|---|
| x | x | 0.274 | 0.027 |
| x |  | 0.260 | 0.001 |
|  | x | **0.296** | **0.740** |

## M.4 Using TEARS Base to Initialize the Text-Encoder

We aim to investigate if pre-initializing the backbone text-encoder as a trained TEARS model is a viable strategy when training aligned models. Table 12 showcases the results of this experiment. Interestingly, we find that pre-initializing the text-encoder does not yield benefits. We hypothesize that the model has learned to map the summary embeddings far away from the black-box embeddings, adding complexity to the optimization process. In comparison, directly training the text-encoder to directly align to the black-box embeddings seems stabilize the training procedure.

**Table 12: Performance metrics of different model configurations.**

| Model | NDCG@50$_c$ | $|\Delta_{\text{up}}@20|$ |
|---|---|---|
| TEARS$_{\text{pre-initialized}}$-MacridVAE | 0.275 | 0.0331 |
| TEARS-MacridVAE | **0.296** | **0.740** |

## M.5 Effect of $\lambda_1$ on Controllability and Recommendation Performance

We analyze the impact of $\lambda_1$, the scaling parameter for the optimal transport loss, has on overall performance and controllability. We take a similar approach to visualizing controllability and display the recommendation performance and controllability of models trained with varying values of $\lambda_1$ over varying $\alpha$.

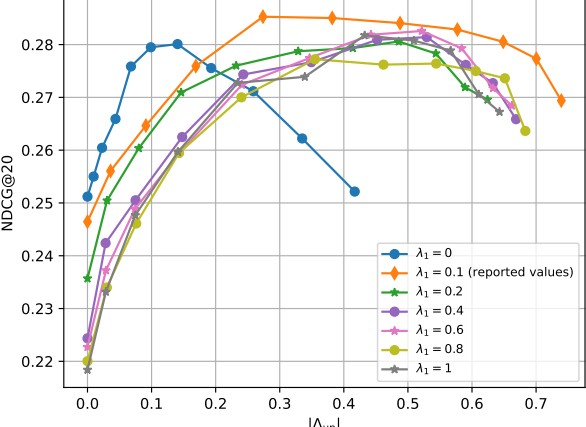

**Figure 12: Visualization of controllability (x-axis) and recommendation performance (y-axis) for varying $\alpha$ (increasing left to right) with models trained with different values of $\lambda_1$.**

We observe that the reported value of $\alpha = 0.1$ achieves the best controllability and recommendation performance. Furthermore, when $\lambda_1 = 0$, which is equivalent to not applying the OT procedure, we see the worst performance, highlighting the importance of the OT procedure. Interestingly, for $\lambda_1 > 0.1$, the performance remains similar, suggesting that fine-tuning this parameter is not crucial, thereby simplifying and making the training of TEARS models more efficient.

## N Stochasticity in GPT Generated Summaries

As mentioned in §3.2, GPT's output is non-determinstic by design. As such, we analyze the effect this has on the performance of TEARS. We generate five summaries using five different seeds and measure the variation on NDCG@20 for the ML-1M and Netflix datasets.

**Table 13: Averaged performance and standard deviations of TEARS models over five different summaries. We observe when $\alpha = 1$ the variation is higher and observe smaller variances when $\alpha = 0.5$.**

|  | ML-1M | |
|---|---|---|
|  | $\alpha = 1$ | $\alpha = 0.5$ |
| TEARS RecVAE | $0.262 \pm 0.004$ | $0.287 \pm 0.002$ |
| TEARS MacridVAE | $0.260 \pm 0.003$ | $0.286 \pm 0.002$ |
| TEARS Multi-VAE | $0.245 \pm 0.002$ | $0.269 \pm 0.002$ |
| TEARS Base | $0.247 \pm 0.004$ | N/A |

Table 13 displays the averaged values and standard deviations of NDCG@20 over the five generated summaries. As can be seen, this has the largest variation when $\alpha = 1$ where TEARS only uses the summary embeddings. Additionally, we observe when $\alpha = 0.5$ we observe much less variation, indicating TEARS can consistently extract important information from the summaries.

## O Cold Start Experiment

We analyze the impact of using varying amounts of input items to determine which users may benefit most from TEARS. To do this, we select users with more than 100 items and generate summaries with different input sizes—10, 25, 50, 75, and 100 items—for each user. We then evaluate performance using NDCG@20 on the items that remain after the initial 100-item selection. This ensures that the evaluation is consistent across all scenarios, with the only variable being the number of input items. Figure 13 illustrates how different input amounts affect NDCG@20. We generally observe that as the number of items increases, so does the recommendation performance for both MacridVAE and TEARS MacridVAE, with both models showing similar results across item counts. In contrast, TEARS Base shows a performance decline as item counts increase. This could be due to the LLM's attention being spread across more items, resulting in vaguer summaries, although further experimentation is needed to confirm this.

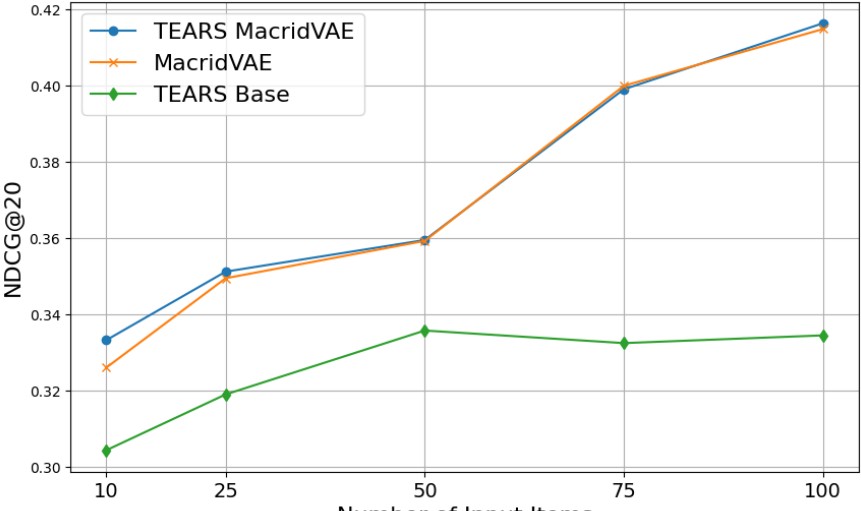

**Figure 13: Plots showcasing the impact of different numbers of input items NDCG@20 for the 78 users in the Netflix dataset. We observe that the information provided by the summaries is specifically important within colder users, while the RecVAE seems to get better the more items that are used. All results for TEARS RecVAE are using the best $\alpha$ according to the validation set.**

## P Generalization to Different LLMs

We aim to explore whether TEARS models can generalize to slightly different writing styles or content than those found in the summaries they were trained on. To assess this, we evaluate the model's recommendation performance using summaries it was not trained on (e.g., evaluating a TEARS model trained on LLaMA summaries with GPT-generated summaries). This will help determine whether there is a significant performance drop when the distribution that generates user summaries is slightly altered. Table 14 presents the results of models trained on GPT and LLaMA summaries and evaluated using various combinations. In general, the models perform best when evaluated with summaries from the same language model they were trained on. However, for the Netflix dataset, interestingly, we observe that the model trained on LLaMA summaries but evaluated with GPT summaries achieves the best performance in both recall@20 and recall@50. Overall, the results suggest that TEARS models demonstrate a strong ability to adapt to different writing styles, as seen in the consistent performance across the datasets. While models tend to perform best when tested with summaries from their training distribution, the performance does not drastically decline when using summaries from other models, indicating flexibility in handling diverse styles and content.

**Table 14: Performance of TEARS models trained on GPT and LLaMA summaries, evaluated across various combinations of training and evaluation datasets. Models perform best when using the summaries of the LLM they were trained on, though the LLaMA-trained model achieves the highest recall@20/50 on GPT-generated summaries for the Netflix dataset. These results highlight TEARS' ability to generalize across different writing styles with minimal performance decline.**

|  | Model Trained & Summaries Used | Recall@20 | NDCG@20 | Recall@50 | NDCG@50 |
|---|---|---|---|---|---|
| ML-1M | LLaMA (LLaMA summaries) | **0.319** ± 0.005 | **0.282** ± 0.005 | 0.363 ± 0.003 | **0.287** ± 0.002 |
|  | LLaMA (GPT summaries) | 0.307 ± 0.004 | 0.271 ± 0.005 | 0.371 ± 0.007 | 0.282 ± 0.004 |
|  | GPT (GPT summaries) | 0.307 ± 0.002 | 0.273 ± 0.002 | **0.374** ± 0.002 | 0.285 ± 0.001 |
|  | GPT (LLaMA summaries) | 0.312 ± 0.003 | 0.277 ± 0.002 | 0.369 ± 0.003 | 0.286 ± 0.002 |
|  | RecVAE | 0.300 ± 0.005 | 0.264 ± 0.003 | 0.360 ± 0.003 | 0.274 ± 0.003 |
| Netflix | LLaMA (LLaMA summaries) | 0.518 ± 0.001 | **0.544** ± 0.001 | 0.457 ± 0.001 | **0.485** ± 0.001 |
|  | LLaMA (GPT summaries) | **0.519** ± 0.001 | **0.544** ± 0.001 | **0.458** ± 0.001 | **0.485** ± 0.001 |
|  | GPT (GPT summaries) | 0.517 ± 0.001 | 0.543 ± 0.000 | 0.457 ± 0.001 | **0.485** ± 0.001 |
|  | GPT (LLaMA summaries) | 0.516 ± 0.001 | 0.542 ± 0.001 | 0.457 ± 0.001 | 0.484 ± 0.001 |
|  | RecVAE | 0.515 ± 0.003 | 0.540 ± 0.003 | 0.455 ± 0.002 | 0.482 ± 0.002 |
| Goodbooks | LLaMA (LLaMA summaries) | 0.173 ± 0.001 | 0.179 ± 0.001 | 0.191 ±0.002 | 0.181 ±0.000 |
|  | LLaMA (GPT summaries) | 0.171 ± 0.001 | 0.177 ± 0.001 | **0.193** ± 0.001 | 0.181 ± 0.001 |
|  | GPT (GPT summaries) | **0.175**± 0.002 | **0.181** ± 0.002 | **0.193** ± 0.000 | **0.183**± 0.001 |
|  | GPT (LLaMA summaries) | 0.172 ± 0.001 | 0.178 ± 0.001 | 0.191 ± 0.002 | 0.180 ± 0.001 |
|  | RecVAE | 0.171 ± 0.001 | 0.176 ± 0.001 | 0.191 ± 0.002 | 0.179 ± 0.001 |

# Q GPT-4-preview Example Summaries

## Q.1 ML-1M

| User ID | Summary |
|---------|---------|
| 2,528 | Summary: The user has a strong preference for crime dramas, often enjoying those that incorporate elements of thrillers and mysteries. They are particularly drawn to complex narratives that involve intricate plots, moral ambiguity, and character-driven stories. The user appreciates the tension and intellectual engagement that comes with unraveling a mystery or following the maneuvers of a criminal mastermind. They also show a high regard for films that blend crime with dramatic depth, exploring the human condition and the consequences of criminal activity. On the other hand, the user does not enjoy crime movies that incorporate excessive comedy or romance, suggesting a preference for more serious and gritty narratives over those that might dilute tension with humor or love stories. While other viewers may find the lighter moments in action-comedy crime films entertaining, this user tends to steer clear of those plot points, favoring a more straightforward and intense viewing experience. |
| 709 | Summary: The user shows a clear preference for horror films that often blend with other genres such as thriller, drama, and romance. They seem to particularly enjoy horror movies that incorporate elements of romance or have a dramatic depth, suggesting a taste for character development and emotional engagement within the horror context. The user also appreciates when horror is mixed with comedy, indicating a fondness for films that balance scares with humor, possibly to lighten the mood or add a satirical edge to the horror genre. In terms of plot points, the user appears to enjoy classic horror tropes and narratives that involve supernatural or monstrous entities, as well as storylines that may include a romantic subplot or a dramatic twist. The user seems to appreciate when horror films explore complex characters or present a unique take on the genre. Conversely, the user does not enjoy certain horror sub-genres, particularly those that may lean heavily into action or sci-fi without substantial horror elements. They also seem to have a distaste for horror films that are perceived as lower quality or that may rely excessively on clichés without offering new or engaging content. Other users may enjoy the adrenaline rush and spectacle of action-packed horror or the imaginative aspects of sci-fi horror, even if these elements are not as appealing to this particular user. They might also find charm in campy or less critically acclaimed horror movies, appreciating them for their cult status or nostalgic value. |
| 3,212 | Summary: The user enjoys comedies that often blend with drama, appreciating films that offer a mix of humor and more serious undertones. They also show a preference for dramas that provide deep, character-driven narratives. The user seems to enjoy plot points that revolve around personal growth, human relationships, and perhaps satirical takes on life and society. On the other hand, the user does not enjoy action-heavy genres, particularly those that involve war themes or are set in science fiction universes. They also seem to have a lower appreciation for romance when it is the central theme of the comedy. Plot points involving high-stakes conflicts, extensive use of special effects, or those that focus on fantastical elements are less appealing to the user. However, these elements may be appreciated by other viewers who enjoy escapism and the thrill of action-packed sequences or the imaginative aspects of science fiction and fantasy. |
| 2,701 | Summary: The user shows a strong preference for action-packed narratives with elements of thriller, drama, and science fiction. They enjoy complex storylines that involve crime-solving, high-stakes scenarios, and intense character-driven plots, often with a psychological or noir twist. The user appreciates when action is blended with deeper themes and when the plot includes unexpected twists or sophisticated narratives that challenge the protagonist both physically and mentally. On the other hand, the user does not favor comedies, particularly those that rely on slapstick humor or light-hearted romantic storylines. They also seem to have less interest in war dramas, despite their appreciation for action and drama in other contexts. Plot points involving straightforward comedy without a substantial or thrilling storyline, or those that lean towards overly sentimental romance, are less enjoyable for the user. However, these elements may be appreciated by other viewers who prefer lighthearted entertainment or are fans of romantic narratives. |

## Q.2 Netflix Summaries

| Netflix ID | Summary |
| --- | --- |
| 2,306,956 | Summary: The user enjoys a diverse range of genres, showing a particular affinity for films that blend action with other elements such as sci-fi, thriller, and crime. They appreciate complex narratives that include mystery and unexpected twists, as well as biographical dramas that offer deep character studies and emotional depth. The user also has a taste for comedies that incorporate elements of drama, romance, and music, suggesting a preference for stories with a rich emotional palette and a balance between humor and heartfelt moments. Additionally, the user is drawn to sports-themed films that likely combine personal growth with the excitement of competition. Conversely, the user does not enjoy certain genres as much, such as pure horror films, which may be due to their often suspenseful and sometimes unsettling nature. Plot points involving supernatural scares or slasher elements are less appealing to the user. While other viewers might find the adrenaline rush and tension of horror thrilling, these aspects do not resonate as strongly with the user's preferences. |
| 1,161,915 | Summary: The user enjoys a blend of genres, with a particular fondness for comedies that intertwine with other genres like crime, romance, and fantasy. They appreciate plot points that involve humorous situations, unexpected romantic developments, and fantastical elements that add a whimsical twist to the narrative. The user also shows a preference for action-packed thrillers, especially those that incorporate elements of adventure, mystery, and science fiction, suggesting a taste for high-stakes scenarios and intricate plotlines. Conversely, the user does not enjoy certain types of comedies, particularly those that may be perceived as lowbrow or lacking in substance. They also seem to steer clear of horror films that lean heavily into the fantasy genre, indicating a disinterest in plot points that combine supernatural elements with horror tropes. While other users may find appeal in the unique blend of comedy and horror or the slapstick nature of certain comedies, these elements do not resonate with the user's preferences. |
| 2,261,374 | Summary: The user enjoys a variety of genres with a strong preference for Crime, Drama, and Documentary films. They appreciate complex narratives that delve into the intricacies of human behavior, moral dilemmas, and social issues. Plot points involving mystery, psychological tension, and character-driven stories seem to resonate well with the user. They also show an interest in films that incorporate historical and biographical elements, suggesting a preference for stories that offer a sense of realism or are grounded in real-world events. Conversely, the user does not enjoy genres that lean heavily on Action and Horror. They seem to be less interested in plot points that prioritize high-octane sequences, gore, or supernatural elements over character development and narrative depth. While other users may find excitement in adrenaline-fueled action scenes or the thrill of horror tropes, these aspects do not align with the user's cinematic tastes, which favor more intellectually stimulating and emotionally rich experiences. |
| 807,353 | Summary: The user enjoys a variety of genres, with a particular affinity for action, drama, and thriller films that often incorporate elements of adventure, crime, history, and war. They appreciate complex narratives that involve high-stakes situations, such as battles, espionage, and survival against overwhelming odds. The user also shows a strong preference for films that delve into historical contexts or speculative futures, often enjoying the interplay between reality and science fiction. Comedies, especially those blended with action or crime, also resonate well, suggesting a taste for humor amidst tension. Conversely, the user does not enjoy certain comedies, particularly those that lean heavily on romance without the balance of another engaging genre. Plot points that revolve solely around romantic entanglements or slapstick humor without a deeper narrative or thematic substance seem to be less appealing. Additionally, dramas that focus primarily on romance or personal turmoil without a broader social or historical context do not capture the user's interest as much. While other users may find these elements relatable or emotionally resonant, they do not align with this user's preferences for complexity and action-oriented storytelling. |
| 769,356 | Summary: The user enjoys a variety of genres, with a particular affinity for action, adventure, thriller, and comedy. They appreciate plot points that involve high-stakes scenarios, such as crime-solving, espionage, and intense physical challenges, often with a blend of humor or romance to balance the tension. The user also shows interest in biographical dramas that tell the stories of remarkable individuals, as well as fantasy elements that add a unique twist to the narrative. Conversely, the user does not enjoy certain romantic comedies and dramas, particularly those that may be perceived as formulaic or lacking in depth. Plot points involving mundane romantic entanglements or overly sentimental narratives are less appealing to the user. While other viewers may find charm and relatability in these stories, the user prefers more dynamic and complex storytelling. Additionally, the user is not fond of certain fantastical musicals or animation that might not align with their taste for more grounded or action-oriented entertainment. |

## Q.3 Goodbooks Summaries

| Goodbooks ID | Summary |
| --- | --- |
| 5,055 | Summary: The user enjoys genres like fantasy, historical fiction, dystopian, and complex emotional narratives. Stories involving time travel, intricate plots with political intrigue, survival against the odds, and magical worlds in which characters undergo significant growth are highly appealing. The user favors narratives with deep character development and rich settings, especially those that combine adventure with moral complexities. The user gravitates towards plot points that feature epic quests, battle between good and evil, elaborate world-building, and cohesive series with consistent character arcs. They appreciate tales of personal sacrifice, love across time, sociopolitical undercurrents, and the fight for justice. The user does not enjoy genres such as romance with superficial or cliched elements, straightforward memoirs, and classic literature with dense language or outdated societal norms. Plot points focusing heavily on melodrama, predictable love triangles, existential navel-gazing, or prolonged internal monologues do not resonate as well with the user compared to tales of adventure and moral conflict. |
| 8,454 | Summary: The user enjoys genres such as classic literature, gothic horror, and fantasy, with a particular fondness for iconic series and novels with rich, atmospheric settings. They appreciate compelling character studies, dark themes, psychological horror, and the use of poetic language. They also seem to enjoy classic sci-fi and dystopian futures, as well as humor and satire presented in graphic novel formats, especially when they exhibit a sharp, witty edge. The user enjoys plot points that delve into the complexity of human nature, transformations or dualities within characters, and epic quests filled with detailed world-building. They favor narratives that are introspective and explore themes of morality, identity, and existentialism, often against a richly described backdrop. Complex and flawed protagonists who evolve through personal conflicts are key elements they appreciate. The user does not seem to enjoy genres like romance or young adult fantasy unless it's integral to a familiar and larger frame, showcased by lukewarm responses to certain popular young adult fantasy books. They also aren't particularly drawn to manga series unless they are critically acclaimed or unique. Heavy reliance on teenage romance, predictable tropes, or overly simplistic narratives particularly deter them. Plot points centered on romantic entanglements or conventional coming-of-age themes do not seem to resonate as well with their tastes. |
| 543 | Summary: The user enjoys genres that blend science fiction, dystopian themes, and intricate horror elements, evident from their high ratings for books involving societal collapse, supernatural occurrences, and deep psychological thrills. Detailed world-building, complex character development, and thought-provoking themes about humanity's survival and moral choices captivate the user. These preferences show in their consistent appreciation for narratives that intertwine existential threats with personal struggles, lush with symbolic and figurative language. The user does not enjoy genres that heavily focus on surreal, darkly comedic or critically nihilistic perspectives, often feeling unsatisfied with their bizarre plot developments and fragmented narrative styles. Books with overly grotesque, chaotic plot lines, and where shock value overshadows substantive storytelling elements, tend to receive lower ratings from the user. Additionally, tales driven by nonsensical or overly convoluted story arcs, without cohesive progression, fail to maintain the user's interest. |
| 7,584 | Summary: The user enjoys genres that span classic literature, fantasy, and magical realism. They have a particular interest in character-driven plots with strong developmental arcs, intricate world-building, and themes of moral complexity. Books like epic fantasies, coming-of-age tales, and those with heavy social commentary resonate well with the user, often featuring a blend of adventure, personal growth, and complex societal interactions. The user appreciates works where characters face profound internal and external conflicts, often set against richly detailed and immersive backdrops. They tend to favor narratives with elements of heroism, complex world settings, and stories that interweave mythological or fantastical elements with deep, layered character dynamics. Conversely, the user does not enjoy genres with straightforward, linear adventure plots or those that heavily focus on satire and political allegory without deeper character engagement. They seem less interested in books where the primary focus is on procedural elements or those centered around detailed explorations of dystopian themes without accompanying rich character development. Additionally, historical fiction and adventures centered in more straightforward and less nuanced narrative structures are generally less appealing to the user. |

# R  LLaMA 3.1 Example Summaries

## R.1  ML-1M

| User ID | Summary |
| --- | --- |
| 2,528 | Summary: The user seems to enjoy crime and thriller movies, particularly those with complex plots and morally ambiguous characters. They appreciate films that explore the darker aspects of human nature, such as violence, corruption, and redemption. The user also enjoys movies with unexpected twists and turns, keeping them engaged and guessing until the end. In terms of specific plot points, the user seems to enjoy stories about heists, gangsters, and law enforcement, as well as films that explore themes of loyalty, betrayal, and revenge. They also appreciate movies with strong, complex characters, particularly those with troubled pasts or conflicted motivations. On the other hand, the user does not seem to enjoy movies that are overly formulaic or predictable, particularly those that rely on tired tropes or cliches. They also tend to dislike films that are overly sentimental or emotionally manipulative, preferring instead movies that are gritty, realistic, and unflinching in their portrayal of the world. In terms of genres, the user does not seem to enjoy horror movies, particularly those that rely on cheap jump scares or gore. They also tend to dislike sci-fi movies that are overly reliant on special effects or convoluted plot twists. However, they do seem to appreciate sci-fi movies that explore complex themes and ideas, particularly those that are grounded in realistic science and technology. |
| 709 | Summary: The user enjoys horror movies with a mix of thriller, comedy, and sci-fi elements. They seem to appreciate classic horror franchises such as Friday the 13th, A Nightmare on Elm Street, and Halloween, as well as more comedic takes on the genre like Ghostbusters and Fright Night. The user also enjoys horror movies with a strong romantic element, such as Bram Stoker's Dracula and Interview with the Vampire. On the other hand, the user does not enjoy horror movies that are too campy or cheesy, such as the Toxic Avenger series or Class of Nuke 'Em High. They also seem to dislike horror movies that rely too heavily on gore or cheap jump scares, such as the Saw or Hostel franchises (although these are not explicitly mentioned in the provided data). The user may also not enjoy horror movies that are too slow-paced or atmospheric, such as The Blair Witch Project. |
| 3,212 | Summary: The user enjoys comedy and drama genres, particularly when they are combined. They seem to appreciate witty dialogue, quirky characters, and offbeat humor, as seen in films with complex, character-driven plots. The user also enjoys romance and comedy when they are intertwined, often with a touch of satire or social commentary. In terms of plot points, the user seems to enjoy stories that explore themes of personal growth, redemption, and self-discovery. They appreciate complex characters with flaws and relatable struggles, as well as narratives that balance humor and pathos. The user also enjoys films with strong ensemble casts and character-driven storytelling. On the other hand, the user does not enjoy action-oriented genres, such as action, thriller, and war films. They seem to find these genres too intense or formulaic, and may not appreciate the emphasis on spectacle over character development. The user also does not enjoy sci-fi and fantasy films, possibly due to their often complex world-building and reliance on special effects. In terms of specific plot points, the user may not enjoy films with overly simplistic or predictable storylines, or those that rely too heavily on convenient plot devices or cliches. They may also not appreciate films with weak character development or those that prioritize style over substance. |
| 2701 | Summary: The user enjoys action-packed movies with thrilling plots, particularly those that combine elements of crime, mystery, and sci-fi. They seem to appreciate films with complex storylines, unexpected twists, and a sense of urgency. Movies with a strong sense of tension and suspense, such as those found in the thriller and film-noir genres, also appeal to the user. On the other hand, the user does not enjoy movies that are overly comedic or lighthearted, particularly those that rely on slapstick humor or clichéd romantic plotlines. They also seem to be less interested in movies that focus on character development or emotional drama, instead preferring films that prioritize plot and action. In terms of specific plot points, the user seems to enjoy movies that feature high-stakes action sequences, intricate heists, and cat-and-mouse chases. They also appreciate films that explore themes of deception, betrayal, and redemption. However, they may not enjoy movies that focus on sentimental relationships, personal growth, or social commentary. |

## R.2 Netflix

| User ID | Summary |
|---|---|
| 2,306,956 | Summary: The user enjoys a wide range of genres, including action, comedy, drama, romance, and thriller. They seem to appreciate films with complex plots, unexpected twists, and strong character development. Movies with a mix of humor, adventure, and heartwarming moments also appeal to them. The user has a soft spot for films based on true stories, biographies, and sports dramas. In terms of specific plot points, the user enjoys stories with redemption arcs, underdog tales, and explorations of social issues. They also appreciate films with clever dialogue, witty banter, and memorable characters. The user is drawn to movies with a strong sense of nostalgia, as well as those that explore themes of love, family, and friendship. On the other hand, the user does not enjoy films that are overly formulaic, predictable, or lacking in depth. They tend to avoid movies with excessive violence, gore, or gratuitous content. The user is also not a fan of films that rely too heavily on slapstick humor or crude comedy. While they appreciate a good horror movie, they tend to prefer those with a more psychological or suspenseful approach rather than outright gore. Overall, the user's tastes are quite eclectic, and they are open to exploring a wide range of genres and themes. However, they tend to prioritize films with strong storytelling, memorable characters, and a mix of humor, heart, and adventure. |
| 1,161,915 | Summary:The user enjoys a wide range of genres, including action, adventure, comedy, crime, drama, fantasy, horror, mystery, romance, sci-fi, and thriller. They seem to appreciate complex plots with unexpected twists and turns, as well as stories that combine elements of multiple genres. The user also enjoys films with strong characters, witty dialogue, and a mix of humor and suspense. In terms of specific plot points, the user seems to enjoy stories that involve adventure, danger, and excitement, such as heists, chases, and battles. They also appreciate films that explore themes of identity, morality, and social commentary, as well as stories that feature clever plot twists and surprises. On the other hand, the user does not enjoy films that are overly simplistic, predictable, or formulaic. They seem to dislike movies that rely too heavily on cliches, stereotypes, or tired genre conventions. The user also appears to be less interested in films that focus primarily on slapstick comedy, gross-out humor, or mindless action. Other users may enjoy films that the user does not, such as those that prioritize style over substance, feature excessive violence or gore, or rely too heavily on special effects. However, the user's tastes suggest that they value intelligence, creativity, and originality in their films, and are willing to explore a wide range of genres and styles in search of something that truly engages and entertains them. |
| 2,261,374 | Summary: The user enjoys crime and drama genres, particularly those with complex plots and moral ambiguity. They appreciate films that explore themes of justice, morality, and the human condition. Movies with intricate storylines, unexpected twists, and strong character development tend to captivate their attention. The user also has a fondness for documentaries, especially those that delve into historical events, social issues, and personal stories. In contrast, the user does not enjoy action-oriented films with simplistic plots and lacking in depth. They tend to dislike movies that prioritize special effects over storytelling and character development. The user is also not fond of romantic comedies or horror movies, finding them to be formulaic and predictable. Furthermore, they are not drawn to films with overtly sentimental or melodramatic tone, instead preferring more nuanced and thought-provoking cinema. |
| 807,353 | Summary: The user enjoys action-packed movies with thrilling plotlines, often featuring crime, adventure, and sci-fi elements. They appreciate films with complex characters, intricate mysteries, and unexpected twists, as seen in genres like mystery, thriller, and crime. The user also has a soft spot for romantic comedies, sports dramas, and historical epics, indicating a appreciation for diverse storytelling. On the other hand, the user does not enjoy movies that are overly reliant on slapstick humor, absurdity, or formulaic rom-com tropes. They tend to dislike films that lack depth, nuance, or originality, often giving lower ratings to movies that feel predictable, shallow, or clichéd. While they may appreciate some animated films, they seem to prefer live-action movies with more mature themes and complex storytelling. Overall, the user values engaging narratives, memorable characters, and cinematic craftsmanship. |
| 769,356 | Summary: The user enjoys action-packed movies with thrilling sequences, often with a mix of adventure, crime, and sci-fi elements. They seem to appreciate films with a strong narrative drive, memorable characters, and impressive stunts. Comedies with a touch of adventure, fantasy, or romance also appeal to them, particularly if they have a unique tone or quirky humor. On the other hand, the user does not enjoy romantic comedies or dramas with overly sentimental or clichéd storylines. They seem to dislike movies that focus too much on relationships or character development at the expense of plot or action. Additionally, they may not appreciate films with a slow pace, convoluted narratives, or an overemphasis on drama or romance. Other users may enjoy these types of movies, but they do not resonate with this particular user. |

## R.3 Goodbooks

| User ID | Summary |
|---------|---------|
| 5,055 | Summary: The user enjoys fiction books that have a strong narrative, well-developed characters, and a mix of romance, drama, and adventure. They seem to appreciate books that explore complex themes, such as love, loss, and self-discovery, and that have a strong emotional resonance. The user also enjoys books that are part of a series, as they allow for a deeper exploration of the characters and their world. In terms of specific genres, the user seems to enjoy young adult fiction, historical fiction, and contemporary fiction. They also appear to have a soft spot for books that are set in the past, particularly in the medieval and Renaissance periods. On the other hand, the user does not enjoy books that are overly simplistic or lack depth. They seem to be turned off by books that are too focused on romance or that have underdeveloped characters. The user also does not enjoy books that are part of the fantasy or science fiction genres, as they seem to prefer more realistic and grounded stories. Overall, the user appears to be a discerning reader who values well-written, engaging stories that explore complex themes and have a strong emotional resonance. |
| 8,454 | Summary: The user enjoys Gothic and horror themes, as well as classic literature and poetry. They seem to appreciate complex, atmospheric storytelling and are drawn to works that explore the human condition, morality, and the supernatural. The user also enjoys science fiction and fantasy, particularly when it involves imaginative world-building and philosophical themes. In terms of plot points, the user seems to enjoy stories that feature mysterious, suspenseful, and introspective narratives, often with a sense of impending doom or catastrophe. They also appreciate stories that explore the nature of good and evil, as well as the consequences of human actions. On the other hand, the user does not enjoy stories that are overly simplistic, clichéd, or lacking in depth. They seem to be turned off by works that rely too heavily on formulaic plot devices or shallow character development. The user also appears to be less interested in stories that are overly focused on action, adventure, or romance, unless these elements are balanced with more substantial themes and character exploration. Overall, the user's tastes suggest a preference for literature that is thought-provoking, emotionally resonant, and intellectually stimulating. |
| 543 | Summary: The user enjoys dystopian and apocalyptic themes, as well as stories that explore the human condition, social commentary, and psychological insights. They seem to appreciate complex, thought-provoking narratives that often feature morally ambiguous characters and blurred lines between good and evil. The user also likes stories that incorporate elements of science fiction, fantasy, and horror, particularly those that involve supernatural or paranormal elements. On the other hand, the user does not enjoy stories that are overly simplistic, predictable, or lacking in depth. They seem to be less interested in tales that focus on straightforward good vs. evil conflicts or those that rely on convenient plot devices. The user also appears to be less enthusiastic about stories that prioritize action and adventure over character development and introspection. Additionally, they may not enjoy narratives that are overly sentimental, clichéd, or reliant on tired tropes. |
| 7,548 | Summary: The user enjoys reading memoirs and biographies, particularly those that focus on overcoming adversity, personal struggles, and inspirational stories. They also appreciate fiction novels that explore complex relationships, family dynamics, and social issues. The user seems to enjoy books that evoke strong emotions, whether it's joy, sadness, or empathy. In terms of plot points, the user likes stories that involve personal growth, transformation, and redemption. They also enjoy narratives that explore themes of love, loss, and self-discovery. The user appears to appreciate books that tackle tough subjects, such as mental health, trauma, and social justice. On the other hand, the user does not enjoy books that are overly simplistic, predictable, or lacking in depth. They seem to prefer stories that are character-driven, nuanced, and thought-provoking. The user may not enjoy books that are too focused on action, adventure, or fantasy, instead preferring more realistic and relatable narratives. Overall, the user's reading preferences suggest a desire for stories that are emotionally resonant, intellectually stimulating, and authentic. They value books that offer a unique perspective, challenge their assumptions, and leave a lasting impact. |

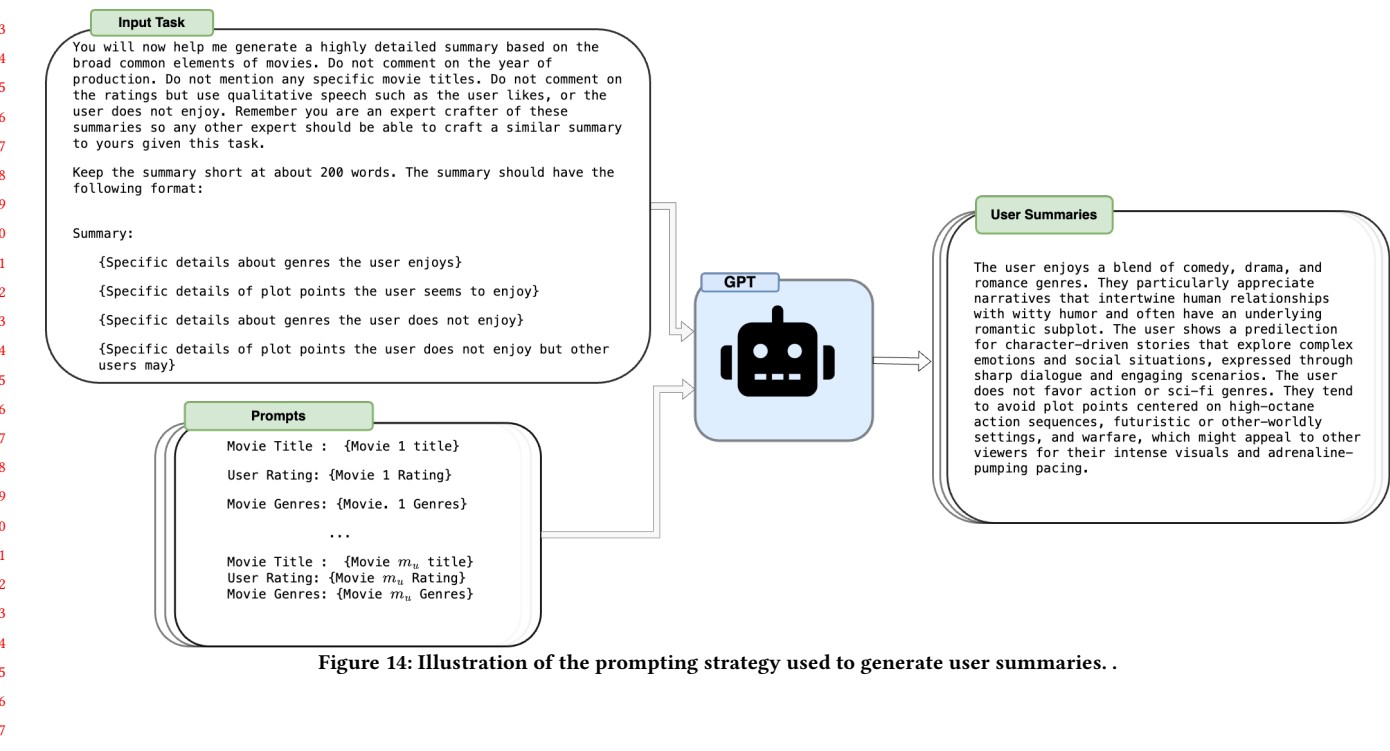

**Figure 14: Illustration of the prompting strategy used to generate user summaries. .**

## R.4 Summary Generation Prompts

| Aspect | Value |
|---|---|
| Prompt for Movies | Task: You will now help me generate a highly detailed summary based on the broad common elements of movies. Do not comment on the year of production. Do not mention any specific movie titles or actors. Do not comment on the ratings but use qualitative speech such as the user likes, or the user does not enjoy. Remember you are an expert crafter of these summaries so any other expert should be able to craft a similar summary to yours given this task. Keep the summary short at about 200 words. The summary should have the following format: 
 Summary: 
 {Specific details about genres the user enjoys}. 
 {Specific details of plot points the user seems to enjoy}. 
 {Specific details about genres the user does not enjoy}. 
 {Specific details of plot points the user does not enjoy but other users may}. 
 {title$_1$}, {rating$_1$}, {genre$_1$} ... {title$_n$}, {rating $_n$}, {genre $_n$ } 
 Do not comment on the ratings or specific titles but use qualitative speech such as the user likes, or the user does not enjoy 
 Do not comment mention any actor names |
| Prompt for Books | Task: You will now help me generate a highly detailed summary based on the broad common elements of books. Do not comment on the year of release. Do not mention any specific book titles or authors. Do not comment on the ratings but use qualitative speech such as the user likes, or the user does not enjoy. Remember you are an expert crafter of these summaries so any other expert should be able to craft a similar summary to yours given this task. Keep the summary short at about 200 words. The summary should have the following format: 
 Summary: 
 {Specific details about genres the user enjoys}. 
 {Specific details of plot points the user seems to enjoy}. 
 {Specific details about genres the user does not enjoy}. 
 {Specific details of plot points the user does not enjoy but other users may}. 
 {title$_1$}, {rating$_1$}, {genre$_1$} ... {title$_n$}, {rating $_n$}, {genre $_n$ } 
 Do not comment on the ratings or specific titles but use qualitative speech such as the user likes, or the user does not enjoy 
 Do not comment or mention any author names |

Received 20 February 2007; revised 12 March 2009; accepted 5 June 2009

