# OpenReview forum: "TEARS: Text Representations for Scrutable Recommendations"
_ACM.org/TheWebConf/2025/Conference — WWW 2025 Poster_

### Official Review · Reviewer_wxyX · 2024-11-16

**Novelty:** 4
**Technical Quality:** 4

**Review:**

This paper introduces TEARS (TExtuAl Representations for Scrutable recommendations), a novel recommender system that combines natural language user summaries with traditional collaborative filtering approaches. Using large language models to generate interpretable user preference summaries and an optimal transport procedure to align them with latent representations, TEARS achieves better performance than standard recommendation models while providing transparency and allowing users to control recommendations through text edits.

**Questions:**

1. Can authors briefly talk about the definition of scrutable recommendations they proposed? It is not a usual term in recommendation area.
2. Is there any technical advantage rather than using LLM? My main concern is in technical novelty, seems authors only use LLM to generate user summary to capture user preference.
3. There has been a lot of work showing that LLM can generate text, but the generated text has many biases. Because LLM uses online resources to generate text, and online resources themselves have a lot of exposure bias or other types of bias, using LLM directly to generate text is not a good approach.

**Reviewer Confidence:**

4: The reviewer is certain that the evaluation is correct and very familiar with the relevant literature

**Scope:**

4: The work is relevant to the Web and to the track, and is of broad interest to the community

---

### Official Review · Reviewer_cbLX · 2024-11-20

**Novelty:** 4
**Technical Quality:** 4

**Review:**

This paper focuses on leveraging LLMs to generate high-quality scrutable user summaries for recommender systems. Optimal transport is employed to align the summaries' representation with the learned representation of a standard VAE for collaborative filtering.

Strengths:
1. I do like the idea of using LLM to summarize and augment user preferences into natural language. Based on the cases shown in Appendix G and P, the user summaries seem quite informative and unique.
2. The plug-and-play TEARS shows improvements across different AE-based models.
3. Ablation study demonstrates the effectiveness of OT.

Weaknesses:
1. In Line 398, it is mentioned that "a maximum of 50 items are considered in the summary." While this is understandable due to LLM input limitations and summary coherence, in real-scenario, it would be more reasonable to construct users' long-term preferences, such as changes in user interests and seasonal variations. I think building long-term user preference trends within this pipeline remains challenging.
2. In Line 455, why was T5-base chosen as the encoder? If the goal here is to obtain better sentence/paragraph representations, why not select commonly used encoder-only LLMs? The authors are suggested to include more discussion about different encoder choices and their impact on overall performance.
3. I'm curious about the efficiency after introducing LLM. There lacks discussion about the overall pipeline efficiency, especially regarding the average time cost of summarization and OT.
4. Have other alignment methods besides OT been tried? For example., contrastive learning. The motivation for using OT is not very clear. Will the model still work if other alignment methods are used?

**Questions:**

See weaknesses.

**Reviewer Confidence:**

3: The reviewer is confident but not certain that the evaluation is correct

**Scope:**

4: The work is relevant to the Web and to the track, and is of broad interest to the community

---

### Official Review · Reviewer_8NCo · 2024-12-01

**Novelty:** 3
**Technical Quality:** 5

**Review:**

Pros:

[Quality] The proposed recommendation system, TEARS, is technically sound. The authors conduct comprehensive evaluations and the results show the superiority of TEARS in comparison to a variety of baselines.

[Clarity] Overall, this manuscript is well presented. The overall flow is easy to follow.

Cons:

[Originality] The originality of this manuscript is limited. The framework of TEARS is, to a great extent, on the basis of the general scrutable recommendations framework proposed by Radlinski et al. [40]. Taking the existing scrutable recommendations framework into consideration, the originality of TEARS is somewhat limited.

In addition, the authors claim that the framework proposed by Radlinski et al. [40] assumes that text summaries can encapsulate all the information typically contained in rich numerical latents, which is generally not the case in practice, potentially leading to a substantial drop in performance. Here, the authors do not provide convincing explanations regarding why the fact that text summaries can encapsulate all the information typically contained in rich numerical latents is generally not the case in practice.

[Significance] The significance of this manuscript is limited as well. Although the proposed model, TEARS, is technically sound, it is unclear how TEARS may improve the users' experience in terms of the recommendation results.

**Questions:**

1. Explain why the fact that text summaries can encapsulate all the information typically contained in rich numerical latents is generally not the case in practice.

2. Provide detailed analysis, empirical study, or both, to demonstrate how TEARS may improve the users' experience in terms of the recommendation results.

**Reviewer Confidence:**

3: The reviewer is confident but not certain that the evaluation is correct

**Scope:**

3: The work is somewhat relevant to the Web and to the track, and is of narrow interest to a sub-community

---

### Official Review · Reviewer_7cjx · 2024-12-01

**Novelty:** 4
**Technical Quality:** 5

**Review:**

## 1. Overview
This paper introduces **TEARS**, a novel recommendation system that represents user preferences through natural language text instead of high-dimensional embeddings, thereby improving recommendation transparency and user control. By leveraging large language models (LLMs) to generate user summaries and employing optimal transport techniques to align text with black-box embeddings, TEARS enhances users' understanding and ability to intervene in recommendation results while maintaining recommendation performance.

## 2. Quality

### 2.1 Strengths
1. The content is very detailed, providing a thorough explanation of the proposed method as well as comprehensive experimental descriptions. However, a significant portion of the content resides in the appendix, which could be compressed in future revisions.
2. The experimental results are comprehensive and convincing.
3. The paper provides implementation code, making the experimental results credible.

### 2.2 Weaknesses
1. The paper is excessively lengthy, which hinders readability. It is recommended to remove some less critical content in future revisions.
2. The experiments are limited to specific datasets (e.g., movies and books) and do not validate the approach in currently trending and highly valuable domains such as short videos and e-commerce.

## 3. Clarity

### 3.1 Strengths
1. The structure of the paper is clear, with strong logical flow from problem definition to method description and experimental analysis.

### 3.2 Weaknesses
1. Figure 2 is not well-aligned with the "Methodology" section. It is recommended to match the module names in the "Methodology" section with those in Figure 2 to improve readability.
2. Certain parts (e.g., the implementation of optimal transport) may require more explanation to enhance comprehensibility.

## 4. Novelty
The combination of **Optimal Transport** and **Convex Combination** in this paper is relatively innovative.

## 5. Significance
By improving the explainability and user control of recommendation systems, this method directly enhances user experience. The approach can be applied to various recommendation scenarios, such as e-commerce and streaming services. However, actual recommendation scenarios are still primarily performance-driven, with limited focus on explainability. Thus, the short-term application potential of this method may be limited.

**Questions:**

1. The paper selected the MovieLens, Netflix, and Goodbooks datasets for experiments. These datasets are commonly used in recommendation system research, but how do you ensure that the experimental results can be generalized to other types of datasets or application scenarios?

2. For long-tail users and items with limited interaction data, the TEARS model may face difficulties in generating summaries. Are there specific strategies to address the long-tail problem?

3. When conducting user summary editing and control experiments, did TEARS consider the varying levels of familiarity users might have with editing operations? Do the experimental results adequately represent the behavior of general users?

**Reviewer Confidence:**

2: The reviewer is willing to defend the evaluation, but it is likely that the reviewer did not understand parts of the paper

**Scope:**

3: The work is somewhat relevant to the Web and to the track, and is of narrow interest to a sub-community

---

### Official Review · Reviewer_561i · 2024-12-02

**Novelty:** 5
**Technical Quality:** 5

**Review:**

## Overview:
The paper presents TEARS (TExtuAl Representations for Scrutable recommendations), addressing the critical challenge of making recommender systems both interpretable and controllable. Traditional collaborative filtering systems rely on opaque high-dimensional embeddings that users cannot understand or meaningfully influence. The authors propose using LLMs to generate natural language summaries of user preferences and develop a hybrid architecture that aligns these summaries with traditional collaborative filtering embeddings using optimal transport techniques.
The authors evaluate TEARS extensively across three datasets (MovieLens-1M, Netflix, and Goodbooks) and demonstrate that it can match or exceed the performance of strong baselines like RecVAE while enabling direct user control. They develop novel evaluation metrics for measuring controllability through large-scope preference changes, fine-grained edits, and guided recommendations.
## Pros:
1. The paper makes a significant technical contribution by thoughtfully combining three key components: modern LLMs for generating interpretable user summaries, optimal transport for aligning text and collaborative filtering embeddings, and a hybrid architecture allowing interpolation between interpretable and black-box recommendations. The ablation studies in Section M clearly demonstrate that each component is essential for the system's success.
2. The authors develop a thorough evaluation methodology addressing both recommendation quality and controllability. This includes standard metrics (NDCG@k, Recall@k) across three diverse datasets, novel controllability metrics for different types of user edits, detailed ablation studies examining architecture choices, and comparisons with both traditional systems and a genre-based interpretable baseline (GERS).
3. The system addresses real limitations of current recommenders while maintaining practicality through several design choices: summaries are constrained to 200 words for readability, the α parameter allows flexible tradeoff between control and performance, the system works with both GPT-4 and open-source LLaMA 3 models, and extensive appendices provide implementation details.
## Cons:
1. While the paper develops automated metrics for controllability, it lacks real user studies examining whether users find the 200-word summaries interpretable, if users can effectively edit summaries to achieve desired changes, and user preferences regarding the interpolation parameter α. The authors acknowledge this limitation in Section 7 but it remains a significant gap.
2. The paper does not thoroughly analyze the computational overhead of generating LLM summaries for all users, updating summaries when users make edits, and training the optimal transport alignment. This analysis is important for understanding real-world scalability.
While the paper provides the prompting strategy in Figure 14, several aspects of summary generation need more analysis. These include consistency of summaries across different LLM runs, variation in summary quality with user history length, and justification for the 200-word length. The brief discussion in Section 3.2 could be expanded.

**Questions:**

Questions *

1. Your evaluation of fine-grained edits in Section 6.2 focuses on items ranked between positions 100-500. How does the system perform for items at different rank ranges? This seems important for understanding the limitations of fine-grained control.
2. The appendix shows some variance in LLM-generated summaries. Have you analyzed how this variance affects recommendation stability? Users might be confused if similar summaries lead to notably different recommendations.
3. Section 3.2 mentions limiting input to 50 items for summary generation. Can you provide more analysis of how summary quality and consistency varies with the number of input items? This seems crucial for understanding scalability to users with longer histories.
4. The optimal transport loss seems central to your method's success (shown in ablations). Could you provide more theoretical analysis of why this alignment works so well compared to alternatives like KL divergence?
5. The comparison with GERS in Section J suggests text summaries provide more value on ML-1M (fewer genres) than Goodbooks (more genres). Could you systematically analyze this relationship between number of genres and relative performance of TEARS vs GERS?

**Reviewer Confidence:**

3: The reviewer is confident but not certain that the evaluation is correct

**Scope:**

4: The work is relevant to the Web and to the track, and is of broad interest to the community